# Interpreting estimated Observation Error Statistics of Weather Radar Measurements using the ICON-LAM-KENDA System

Yuefei Zeng[1], Tijana Janjic[1], Yuxuan Feng[1,2], Ulrich Blahak[3], Alberto de Lozar[3], Elisabeth Bauernschubert[3], Klaus Stephan[3], and Jinzhong Min[2]

[1]Meteorologisches Institut, Ludwig-Maximilians-Universität (LMU) München, Munich, Germany
[2]Key Laboratory of Meteorological Disaster of Ministry of Education/Collaborative Innovation Center on Forecast and Evaluation of Meteorological Disasters, Nanjing University of Information Science & Technology, Nanjing, China
[3]Deutscher Wetterdienst, Offenbach, Germany

**Correspondence:** Yuxuan.Feng@physik.uni-muenchen.de

**Abstract.**

Assimilation of weather radar measurements including radar reflectivity and radial wind data has been operational at the Deutscher Wetterdienst, with a diagonal observation error (OE) covariance matrix. For an implementation of a full OE covariance matrix, the statistics of the OE have to be a priori estimated, for which the Desroziers method has been often used. However, the resulted statistics consists of contributions from different error sources and are difficult to interpret. In this work, we use an approach that is based on samples for truncation error in radar observation space to approximate the representation error due to unresolved scales and processes (RE) and compare its statistics with the OE statistics estimated by the Desroziers method. It is found that the statistics of the RE help the understanding of several important features in the variances and correlation length scales of the OE for both reflectivity and radial wind data and the other error sources from the microphysical scheme, radar observation operator and the superobbing technique may also contribute, for instance, to differences among different elevations and observation types. The statistics presented here can serve as a guideline for selecting which observations are assimilated and for assignment of the OE covariance matrix that can be diagonal or full and correlated.

## 1 Introduction

Nowadays, assimilation of weather radar measurements has been widely adopted in many weather services for convective-scale numerical weather prediction (NWP) models (Gustafsson et al., 2018). For instance, in the 3D-VAR system of the Meteo-France, Doppler radial wind measurements are assimilated (Montmerle and Faccani, 2009), and radar reflectivity measurements are assimilated by a 1D+3D-VAR method (Caumont et al., 2010), which firstly derives relative humidity profiles from reflectivity data. At the Met Office, volume scans of radar reflectivity data are directly assimilated (Hawkness-Smith and Simonin, 2021) by the hourly cycling 4D-VAR (Milan et al., 2020). At the Deutscher Wetterdienst (DWD), the Kilometre-scale ENsemble Data Assimilation (KENDA) system (Schraff et al., 2016) has been developed for the the COSMO (COnsortium for Small-scale MOdelling, Baldlauf et al. 2011) and the ICON (ICOsahedral Nonhydtostatic, Zängl et al. 2015) models. Since June 2020, the radial wind and reflectivity data have been assimilated via the Local Ensemble Transform Kalman Filter

(LETKF, Hunt et al. 2007) combined with the latent heat nudging (Stephan et al., 2008) for the COSMO-model in the operational suite; The ICON-LAM (ICON-Limited Area Model) is the limited area version of the ICON model and is to replace the COSMO model in the operational forecasting system. The ICON-D2 (D: Deutshcland (Germany); 2: 2 km) is an ICON-LAM setting at approximately 2 km grid spacing, which is restricted to Germany and the neighboring countries and became operational for very short-range forecasting since February 2021. Despite of rapid progress, convective-scale data assimilation is still at an early phase of development and a number of challenges remain for both variational and ensemble-based methods, e.g., imbalance due to rapid update (Bick et al., 2016; Lange et al., 2017; Zeng et al., 2021b), strong nonlinearity of models and observation operators (Wang and Wang, 2017), model error due to unresolved scales (Zeng et al., 2019, 2020) and parameters (Ruckstuhl and Janjić, 2020), representation error of observations (Janjic et al., 2018), and etc. In the present work, we focus on the last topic.

As stated in Janjic et al. (2018), the observation error consists of two components in the context of data assimilation: first, the instrument error that occurs during the measurement process; second, the representation error that is understood as the difference between the actual observation and its modelled representation and it can be primarily categorized into three types: observation operator error, pre-processing or quality control error and error due to unresolved scales and processes. In this work, for brevity of text and convenience of explanation, we denote the observation error with "OE" and the instrument error with "IE", and we group the observation operator error together with pre-processing or quality control error as forward model error and denote it with "FE", and denote the representation error due to unresolved scales and processes with "RE" (i.e., OE = IE + FE + RE). In general, the FE and the RE are larger than the IE and the IE is better understood (e.g., standard deviations of the IE for radar reflectivity observations are proportional to the measured values, Doviak and Zrnic 1993; Xue et al. 2007). To quantify the OE statistics, the methods of Hollingsworth and Lonnberg (1986) and Desroziers et al. (2005) have been widely used in practice. The former one is based on the first guess departure, while the latter one is based on the first guess and analysis departures and enjoys more popularity in recent years. For instance, the Met Office uses the Desroziers method to calculate the interchannel error covariances for satellites and incorporates them in the OE covariance matrix in the 3D-VAR (Weston et al., 2014; Waller et al., 2016a), and so does the ECMWF (Bormann et al., 2016). The DWD specifies the OE variances for conventional observations (Schraff et al., 2016) and MODE-S observations (Lange and Janjić, 2016) based on the Desroziers diagnositic in the KENDA system. Furthermore, the Meteo-France, Met Office and JMA (Japan Meteorological Agency) have also applied the method for radial wind observations to estimate spatial error correlations that are then accounted for in the data assimilation (Wattrelot et al., 2012; Simonin et al., 2019; Fujita et al., 2020). In the present work, we use the Desroziers method to explore characteristics of the OE for reflectivity and radial wind in the operational ICON-LAM-KENDA system of the DWD. It is the first application of radar data assimilation using this framework (a similar study has been done by Waller et al. (2019) but for the COSMO-KENDA system and only for the radial wind). To authors' knowledge, it is also the first in-depth attempt to investigate the OE statistics (variances and correlations) of reflectivity data. However, the estimated OE statistics embraces contributions from the IE, FE and RE and it is not clear how much an individual error contributes. To approximate the RE, we assume that a high resolution model is the truth and we regard model equivalence of radar data calculated from the truth as observations (e.g., Waller et al. 2014, 2021) and evaluate the statistics from a set of samples of

differences between observations and model equivalence of the low resolution model run, which can then be compared with the OE statistics estimated by the Desroziers method.

The paper is organized as follows. Section 2 describes the concepts of the two methods used here to compute the observation error statistics. Section 3 gives details about the ICON model and the radar observation operator. Section 4 presents the experimental settings and results, followed by Section 5 for summary.

## 2    Methodology

In this section, we describe two methods used for calculating statistics of the RE and OE.

### 2.1    Samples of error due to unresolved scales and processes

In spite of increasing resolution in operational NWP models, convection can not be completely resolved and shallow convection has to be parametrized. It is known that with a higher horizontal resolution the model can better resolve updraft and vertical transportation of energy and more accurately describe orography (Wedi, 2014). To mimic the RE, one can treat the mapping of states from a high resolution model as observations (Waller et al., 2014), and the low resolution model is considered as a
truncation.

Following the similar approach of Zeng et al. (2019), differences between forecasts of two model runs with different resolutions, expressed in the observation space, are used to represent the RE:

$$\boldsymbol{\eta}_k = \mathcal{H}\left\{\left[\mathcal{M}^H\left(\boldsymbol{x}^H\left(t_k - t\right)\right)\right]\right\} - \mathcal{H}\left\{\mathcal{M}^L\left[\mathcal{T}\left(\boldsymbol{x}^H\left(t_k - t\right)\right)\right]\right\},\tag{1}$$

where $\mathcal{H}$ is the observation operator, $\mathcal{M}^H$ and $\mathcal{M}^L$ are models at high and low resolutions, respectively, $\boldsymbol{x}^H$ is the state of $\mathcal{M}^H$, $\mathcal{T}$ is the interpolation operator. $t$ is the pre-defined forecast time and $t_k$ is an arbitrary valid time. For any $t_k$, we can
calculate a $\boldsymbol{\eta}_k$ that is a sample for the RE. A flowchart of this approach is given in Figure 2 of Zeng et al. (2019).

Running models for a period (with a certain weather pattern), a set of samples are produced. If the size of samples are sufficiently large, statistics of samples should provide useful information on the nature of the RE (under certain weather conditions). More details about the settings of model runs can be found in Section 4.1.

### 2.2    The Desroziers method

The Desroziers method (Desroziers et al., 2005) calculates the expected value of the outer product of the first guess departure (or called innovation) $\mathbf{d}_{o-b} = \mathbf{y}^{\mathbf{o}} - \mathcal{H}(\mathbf{x}_b)$ and the analysis departure $\mathbf{d}_{o-a} = \mathbf{y}^{\mathbf{o}} - \mathcal{H}(\mathbf{x}_a)$ to approximate the observation error covariance matrix:

$$\mathbf{R}_{\text{est}} = E[\mathbf{d}_{o-a}\mathbf{d}_{o-b}^T],\tag{2}$$

where $\mathbf{y}^{\mathbf{o}}$ is the observation vector. $\mathbf{R}_{\text{est}}$ is optimal in case of a linear observation operator and uncorrelated background error
and OE covariances (denoted by $\mathbf{P}^b$ and $\mathbf{R}$) that are perfectly specified (Reichle et al., 2002). Although these assumptions are

usually not satisfied in practice, $\mathbf{R}_{est}$ is still widely used as a qualitative indicator for the OE statistics. Besides, Desroziers et al. (2005) initially suggested applying Eq. 2 in successive iterations to converge to the truth, however, a useful estimation can be often obtained in the first iteration (Waller et al., 2016b). Therefore, considering the computational cost, most of studies with operational NWP models have performed only the first iteration (e.g., Weston et al. 2014; Lange and Janjić 2016; Waller et al. 2016a; Bormann et al. 2016). In this work, we also compute the Desroziers diagnostic in one iteration. Furthermore, as in Waller et al. (2016a), the means of $\mathbf{d}_{o-a}$ and $\mathbf{d}_{o-b}$ are subtracted to ensure that the bias does not affect $\mathbf{R}_{est}$.

In the following, we estimate statistics of the RE of radar reflectivity and radial wind data by using the method from Section 2.1 and statistics of the OE by applying the Desroziers method to an data assimilation experiment with a low resolution model, i.e., $\mathbf{d}_{o-b} = \mathbf{y^o} - \mathcal{H}(\mathbf{x}_b^L)$ and $\mathbf{d}_{o-a} = \mathbf{y^o} - \mathcal{H}(\mathbf{x}_a^L)$. It should be mentioned that due to the logarithmic unit of reflectivity it is very well established in radar data assimilation community to set a threshold value for very small reflectivities (e.g., with negative values) to avoid unrealistically large increments (Zeng et al., 2021a) and spurious convection (Aksoy et al., 2009). In the operational settings of KENDA, the threshold value is 0 dBZ, which means all reflectivity values lower than 0 dBZ are set to 0 dBZ, and we call 0 dBZ data as "clear-air reflectivity data". Before the superobbing, the same threshold value is set to all observations and to all simulated reflectivities in each background ensemble member. However, regarding the Desroziers diagnostics, the standard deviations of the OE may be underestimated since the same threshold value is set to both observations and backgrounds (Zeng et al., 2021a). To mitigate this problem, we calculate Desroziers diagnostics for reflectivities with values $\geq 5$ dBZ either in observations or in backgrounds or in analyses. To be consistent, we also calculate the statistics of the RE for reflectivity data $\geq 0$ dBZ and $\geq 5$ dBZ, respectively.

## 3 The ICON model, radar observations and the observation operator

The ICON (ICOsahedral Nonhydrostatic) global model has been in operation at the DWD since January 2015 (Zängl et al., 2015), which is non-hydrostatic and is based on an icosahedral (triangular) grid with a horizontal resolution of 13 km and 90 vertical levels. The ICON-LAM is the regional model and the ICON-D2 is one version of the ICON-LAM, with domain as shown in Fig. 1 and with a horizontal resolution of 2.1 km and 65 vertical levels. The ICON-D2 model became operational at the DWD since February 2021. Lateral boundary conditions for the ICON-LAM Ensemble Prediction System (EPS) are provided by the global ICON EPS, with a resolution of 40 (20) km globally and 13 (6.5) km over Europe for the ensemble (deterministic run). The deep convection is explicitly resolved and the shallow convection is parametrized with the Tiedtke scheme (Tiedtke, 1989). The turbulent kinetic energy (TKE) scheme for turbulence is developed by Raschendorfer (2001). The Lin-Farley-Orville-type one-moment bulk microphysical scheme is used, which predicts cloud droplets $q_c$, cloud ice $q_i$, rain $q_r$, snow $q_s$ and graupel $q_g$ (Lin et al., 1983; Reinhardt and Seifert, 2006).

The DWD utilizes a network of 17 C-band Doppler radars covering Germany and part of adjacent countries (see Fig. 1), A complete radar volume scan lasts 5 min and it consists of 180 range bins (resolution of 1.0 km), 360 azimuths (resolution of 1.0°) and 10 elevations (0.5°, 1.5°, 2.5°, 3.5°, 4.5°, 5.5°, 8.0°, 12.0°, 17.0°, and 25.0°). To transform model variables to synthetic radar observations, an Efficient Modular VOlume scanning RADar Operator (EMVORADO, Zeng 2013; Zeng

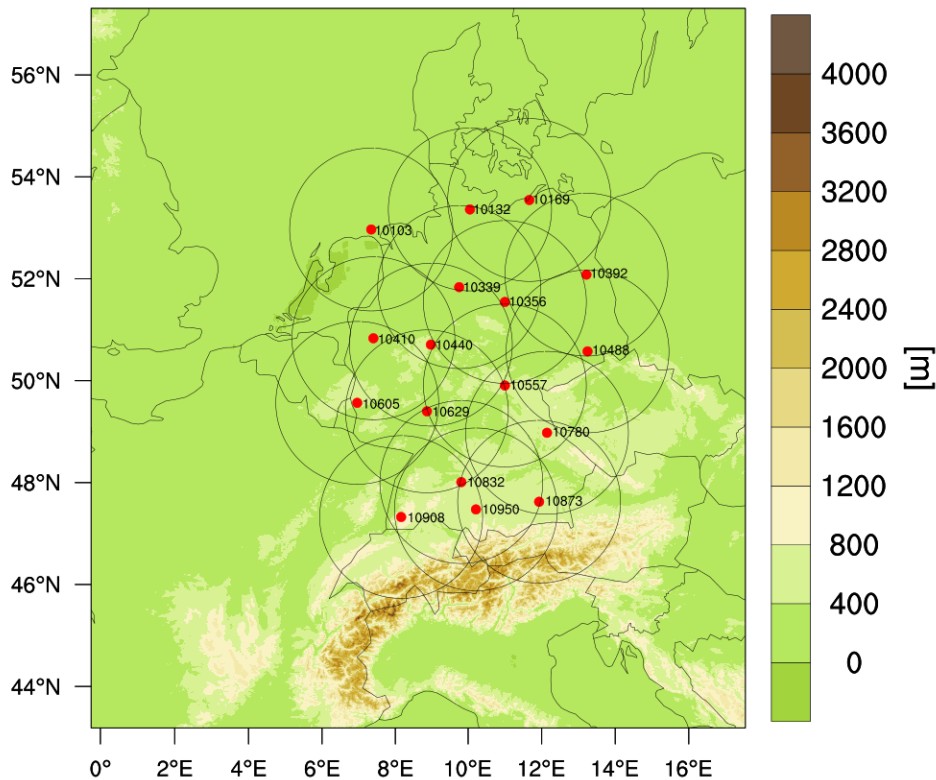

**Figure 1.** Illustration of the ICON-D2 domain with the orography and of the radar network of the DWD (each station is denoted by a red bullet and the station number, the scanning range is denoted by a circle)

et al. 2014, 2016) has been developed. The EMVORADO is coded in a modular way and is able to simulate effects such as
beam bending/broadening/shielding, fall speed and reflectivity weighting for radial wind, attenuation, detectable signal and etc. Reflectivities are first calculated on the model grid points and then interpolated onto radar coordinates. There are two scattering schemes are implemented: the Rayleigh approximation for simple near-spherical hydrometeors whose size is small compared to the wavelength and the Mie method for one- and two-layered spherical hydrometeors of arbitrary size. To simulate radial wind, three wind components are interpolated onto radar coordinates and then radial winds are calculated. In the operational
settings, the EMVORADO is run with the Mie method (using look-up tables) and takes beam shielding, fall speed, attenuation and detectable signal into account. Beam bending and broadening effects as well as reflectivity weighting are omitted for the sake of efficiency (computational costs can be found in Zeng et al. 2016). The 4/3 Earth radius model that assumes a standard atmosphere is used to mimic the beam propagation (Zeng et al., 2014).

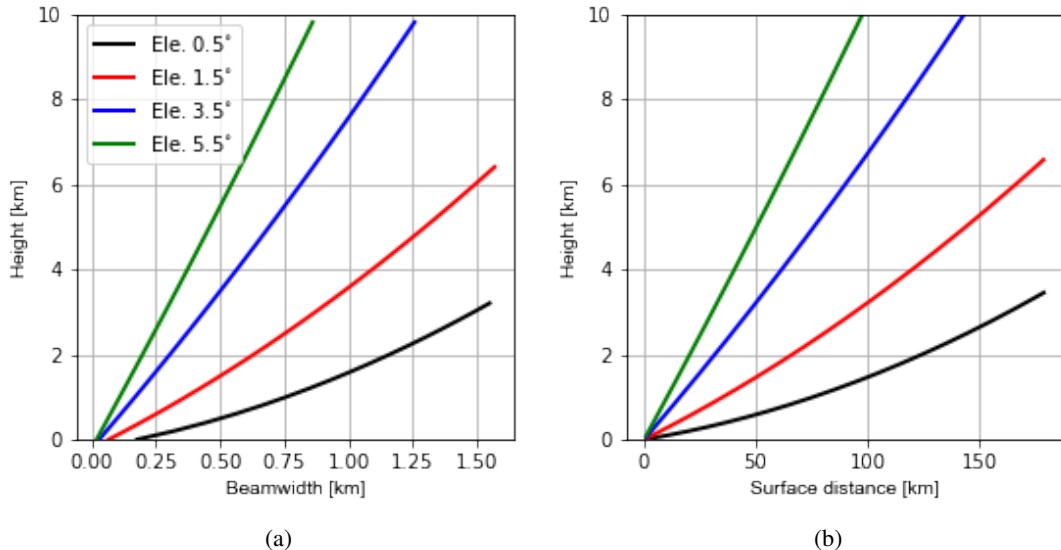

(a)                                                                                    (b)

**Figure 2.** Illustration of variations of beamwidths (in km for azimuth resolution of $1.0°$) with height (left) and surface distances (away from the radar station) with corresponding heights (right) for elevations $0.5°$, $1.5°$, $3.5°$ and $5.5°$, based on the 4/3 Earth radius model. The height of the radar station is omitted. The figure for radial ranges with corresponding heights looks very similar to the right panel.

## 4   Experimental settings and results

In this section, experiments are performed to create samples for estimation of the OE statistics of radar observations. For each elevation, standard deviations and horizontal correlations of the OE at different heights are calculated as in Waller et al. (2016c, 2019). For comparison, the same is done for the RE. Results are shown for elevations $0.5°$, $1.5°$, $3.5°$ and $5.5°$. The elevations higher than $5.5°$ are not shown due to small numbers of samples. As in Liu and Rabier (2002); Waller et al. (2016c, 2019), the correlation length scale is determined by the distance, at which the correlation coefficient is not longer greater than 0.2.

Standard deviations are averaged over all samples and correlations are calculated for each elevation at each radar station for a specific time and then averaged. As Waller et al. (2019), if the numbers of samples available for estimation are too small (e.g., $< 1000$), the estimated standard deviations and correlations might be considerably contaminated by the sampling error and therefore are not reliable. Furthermore, it is noted in Waller et al. (2017) that for the local ensemble data assimilation scheme the error correlation between two observations $y_i$ and $y_j$ estimated by the Desroziers method is correct if the observation

operator applied to calculate the model counterpart of $y_i$ acts only on states updated by $y_j$, however, the LETKF does not seem to suffer strongly from this issue as shown in Waller et al. (2019).

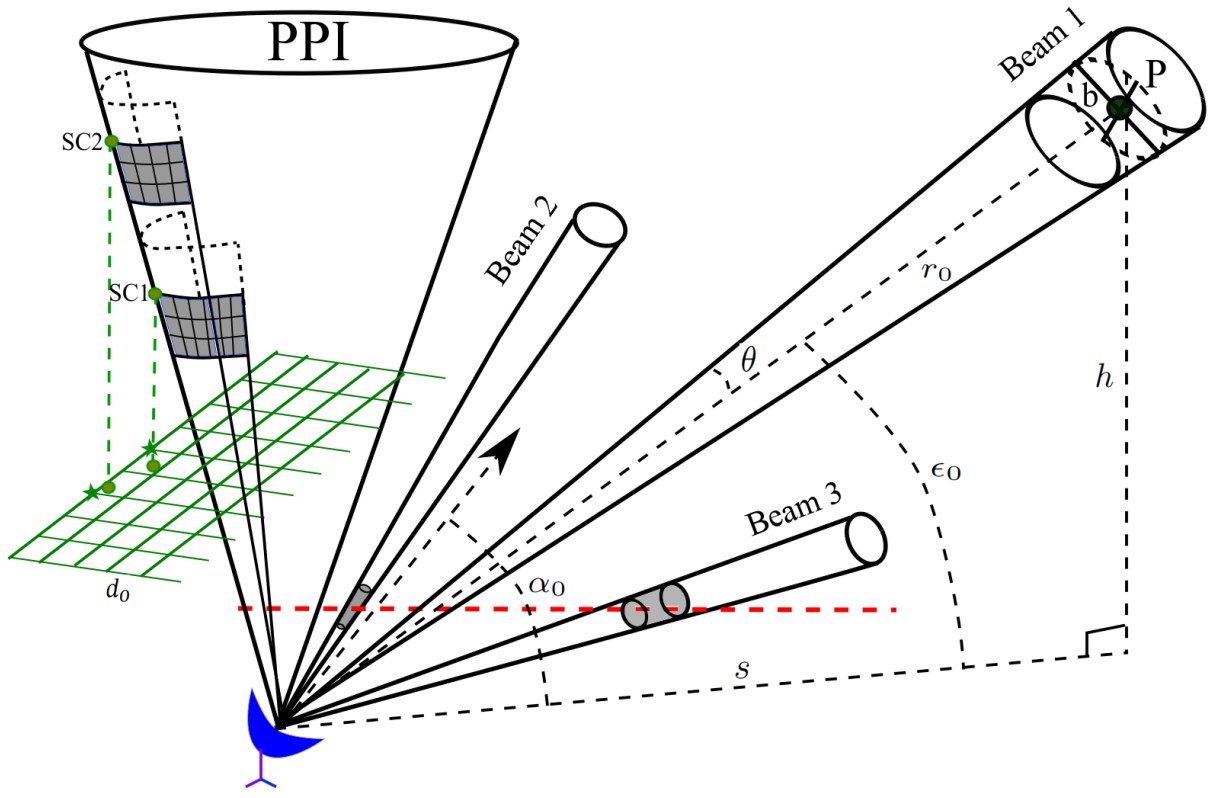

**Figure 3.** Illustration of the radar coordinate $(r, \alpha, \epsilon)$, the beam broadening effect and the superobbing technique: $r_0$, $\alpha_0$ and $\epsilon_0$ are the radial range, azimuth and elevation of the beam 1 at the pulse volume whose center is at the point P (denoted by black bullet). $h$ and $s$ are the height and the surface distance of P (note that the curvature of the Earth is omitted in this figure but not in calculations, e.g., in Fig. 2). $\theta$ and $b$ are beamwidths in degree and km, respectively. Beam 2 has a higher elevation than Beam 3, the pulse volumes (denoted by grey cylinder) from Beam 2 and Beam 3 are at the same height (dented by red dashed line) but the former one is smaller. The superobbing is done for each PPI scan, individually, for which a Cartesian grid (denoted by green solid line) with a fixed resolution $d_o$ is defined. For each grid point, a point from the PPI scan is searched, whose projection is closest to it. Once it is found, e.g., the point denoted by "SC1", a wedge around it is defined, the grey area is the lower right part of the wedge and the grids denote pulse volumes. SC2 is further in the radial range than SC1 and its superobbing wedge contains fewer pulse volumes.

## 4.1 Observation error statistics estimated by samples of error due unresolved scales and processes

### 4.1.1 Experimental settings

To create samples for the RE, the ICON-D2 model (equipped with the EMVORADO) is run with a resolution of 1.0 km for a training period from May 26, 2016, 00:00 UTC to June 25, 2016, 00:00 UTC, which has been investigated in a number of studies (Zeng et al., 2018, 2019, 2020). During the period, a large area of Southeastern and Central Europe was hit by severe thunderstorms with heavy rain. The hourly outputs of the model run at 1.0 km are interpolated onto a coarser grid with 2.1 km (operational) using the iconremap utility from the DWD ICON Tools (Prill, 2014), and the interpolated states are used as initial conditions for 1-h forecast runs at the resolution of 2.1 km. Both high and low resolution model runs are driven by hourly boundary conditions. For any time during this period, one can build a difference between two model runs. In total, there are 720 samples of differences. Since the EMVORADO is run together, we have the samples also in radar observation space. Each sample contains differences in radar volume scans of all radar stations. No superobbing has been applied, and no data assimilation has been conducted here since we are interested in the climatology of the RE instead of exact positions and intensities of convection.

It is recognized in Fig. 2a that the resolution of 1 km is still much coarser than the radar beamwidth (although the difference narrows with increasing height), the RE will be probably underestimated. Nevertheless, the statistics can still provide valuable insights into features of the RE as shown in the following.

### 4.1.2 Results

Fig. 4a shows vertical profiles of the estimated standard deviations of the RE for reflectivity data $\geq 0$ dBZ at elevations $0.5°$, $1.5°$, $3.5°$ and $5.5°$. The biases of the RE are approximately zero (not shown). The numbers of samples used vary from $10^4 \sim 10^6$ near the surface to $\sim 10^4$ at the top (see Fig. A1). Standard deviations for all elevations are similar, which increase till the height of 3 km and then decrease. It is noticed that variations of standard deviations are very comparable to those of simulated reflectivities of the (high resolution) model run. By comparing simulated reflectivities in Fig. 4b with the vertical profiles of the mean states of mixing ratio variables in Fig. 5, it can be deduced that reflectivities below the height of 3 km are mainly attributed to the rain while reflectivities above 3 km are mainly attributed to the graupel. For reflectivity data $\geq 5$ dBZ as shown in Fig. 4e, standard deviations are also very similar for all elevations except at the top, where the higher elevations exhibit slightly smaller errors. Standard deviations increase in the first few hundred meters and then slightly decrease for the next few hundred meters before increasing to a local maximum at around 3 km. Above 3 km, standard deviations decrease till 5 km and then increase to the top. The variations of simulated reflectivities of the model run exhibit a similar pattern although the decrease between 2 and 6 km is sharper. Overall, it can be said that standard deviations of the RE are approximately proportional to true values. Moreover, to see the errors decoupled from observed values (this may be interesting for comparison among elevations because different elevations scan different parts of atmosphere, their measurements may be associated with different errors), we divide standard deviations by simulated data from the high resolution model run that represents observations, this is equal

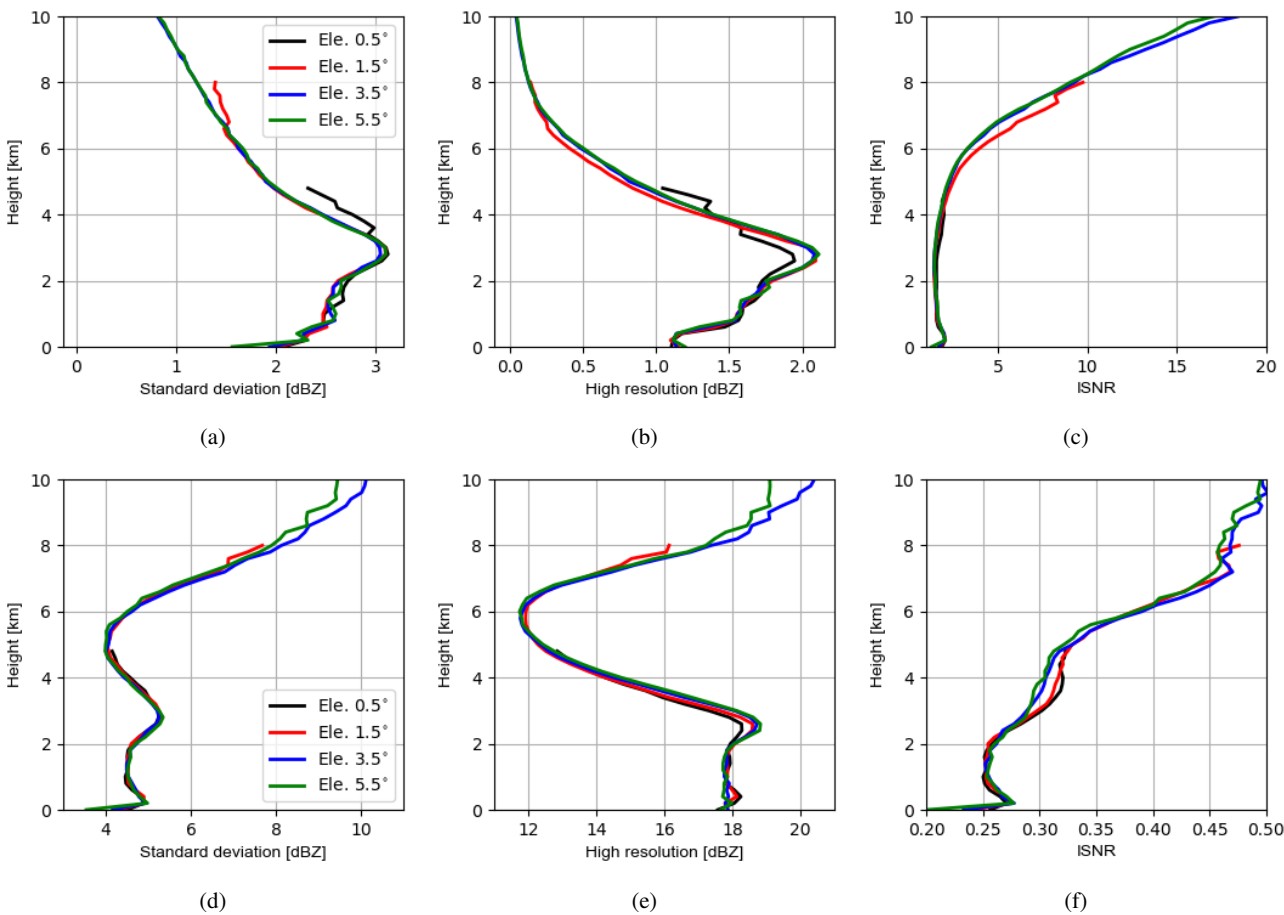

**Figure 4.** a) Vertical profiles of the estimated standard deviations of the RE for reflectivity data $\geq 0$ dBZ at elevations $0.5°$, $1.5°$, $3.5°$ and $5.5°$ b) Vertical profiles of simulated reflectivity data $\geq 0$ dBZ in high resolution model run, averaged over all samples; c) ISNR (inverse of signal-to-noise ratio), i.e, standard deviations in a) are divided by high resolution data in b) for each elevation, respectively; (d-f) for reflectivity data $\geq 5$ dBZ. For each elevation, the samples are binned every 200 m in the vertical. Here we show results only up to 10 km to be consistent with Section 4.2.

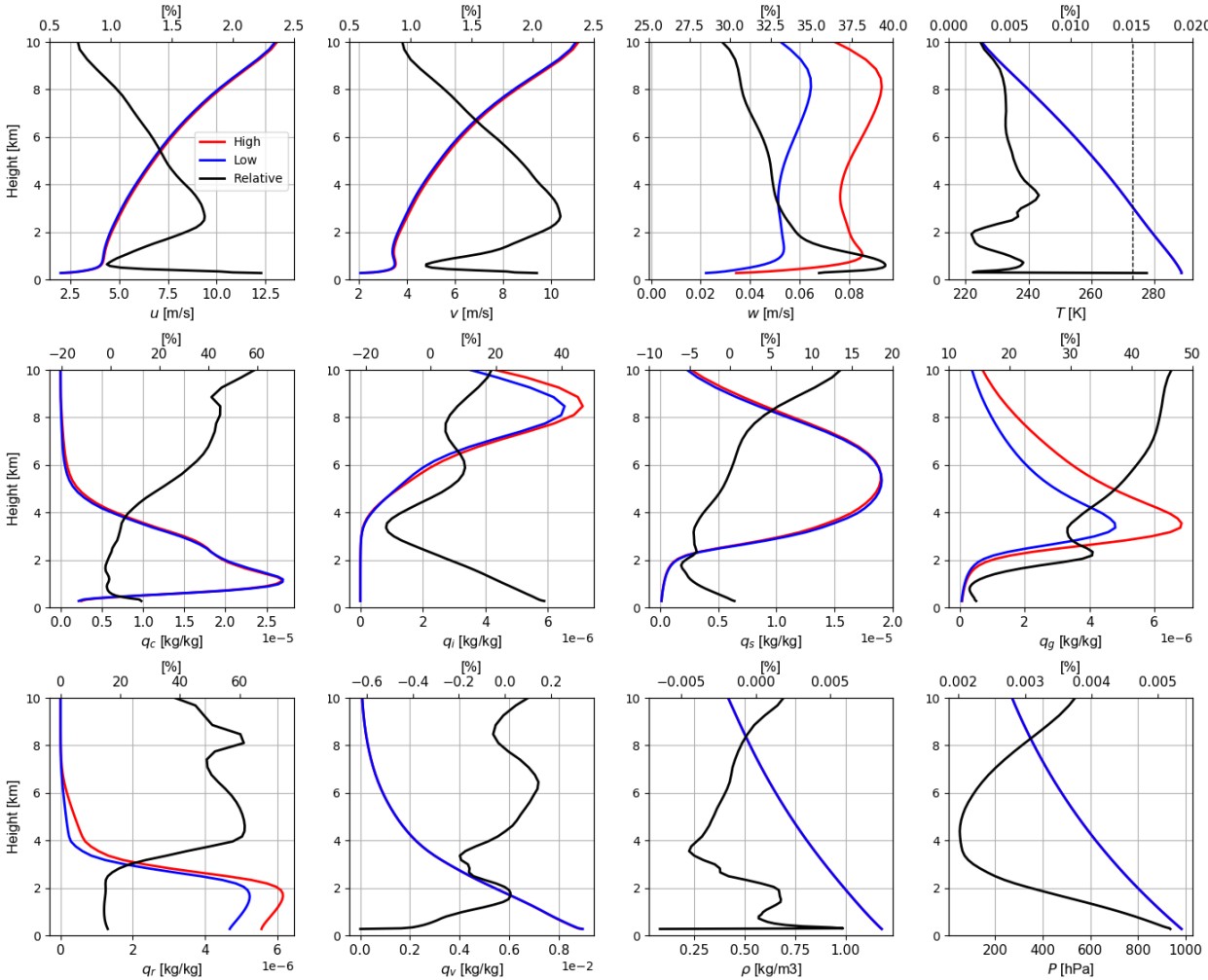

**Figure 5.** Vertical profiles of model variables such as horizontal wind $(u, v)$, vertical wind $w$, temperature $T$, mixing ratios of cloud water $q_c$, cloud ice $q_i$, snow $q_s$, graupel $q_g$, rain $q_r$ and water vapor $q_v$ as well as air density $\rho$ and pressure $P$ in high (1.0 km) and low resolution (2.1 km) model runs. The absolute values of model variables averaged over the entire period are shown. Black vertical dashed line in Figure for $T$ indicates 273 K. Black solid line is the relative difference [%] (the upper x-axis).

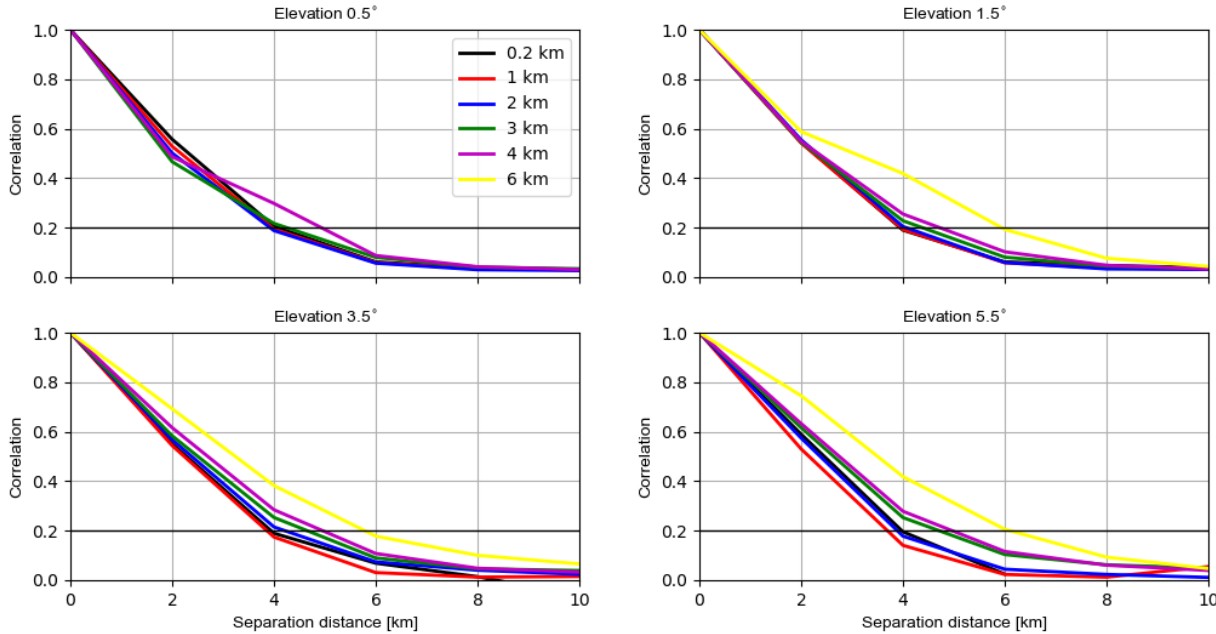

**Figure 6.** The estimated horizontal correlations of the RE for reflectivity data $\geq 5$ dBZ for elevations $0.5°$, $1.5°$, $3.5°$ and $5.5°$ at heights of 0.2, 1.0, 2.0, 3.0, 4.0 and 6.0 km. For each elevation, the samples are binned every 2 km in the separation distance. It is possible for each elevation that samples are not available for some heights.

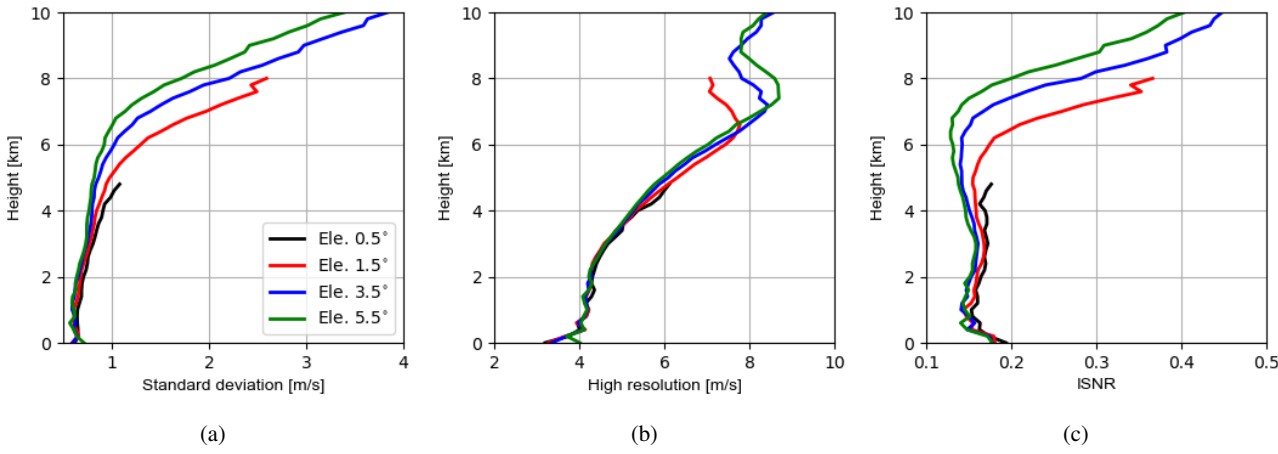

(a)             (b)             (c)

**Figure 7.** The same as Fig. 4 but for radial wind

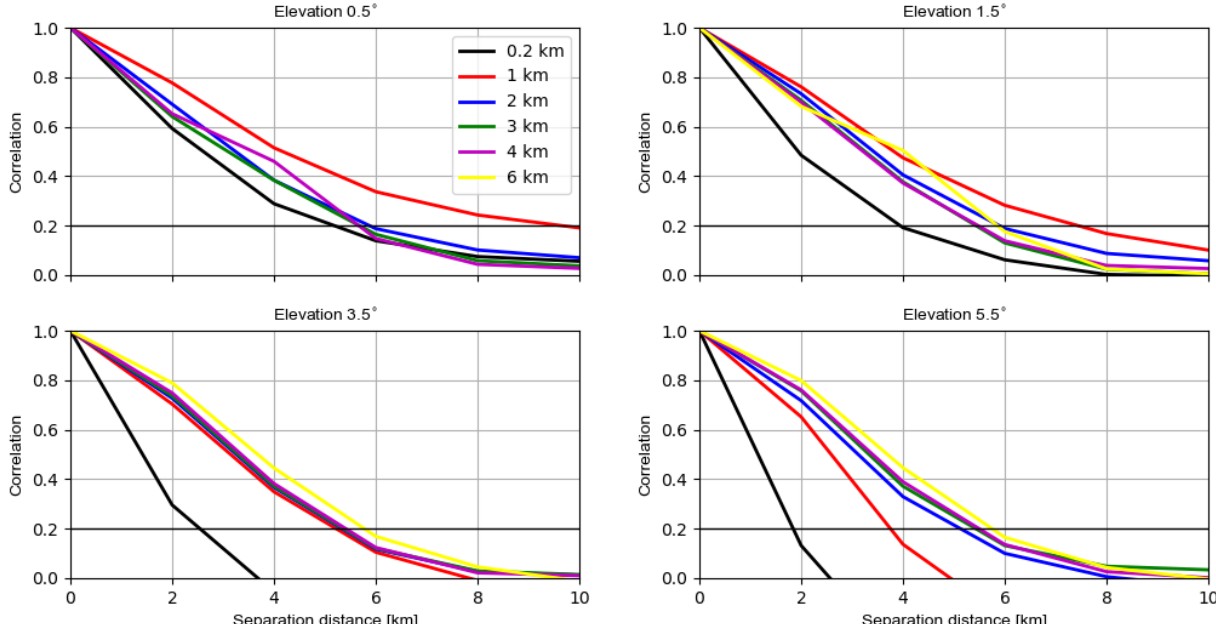

**Figure 8.** The same as Fig. 6 but for radial wind

to the inverse of signal-to-noise ratio (e.g., Russ 2006, hereafter ISNR). As shown in Figs. 4c and 4f, the ISNRs are similar
175    among elevations and they increase consistently except at very low heights.

Fig. 6 shows the horizontal correlations of the RE for reflectivity data $\geq 5$ dBZ (the correlations for $\geq 0$ dBZ are very similar,
not shown). The numbers of samples used for each separate distance and for each elevation are about $10^6 \sim 10^8$ (see Fig. A2).
For the elevation $0.5°$, the correlation length scales at heights of 0.2, 1, 2, 3 and 4 km are given, which are about 4, 4, 4, 4 and
5 km. For $1.5°$, the correlation length scales at heights of 0.2, 1, 2, 3, 4 and 6 km are given, increasing successively from 4 to
180    6 km. The correlation length scales for the other higher elevations look similar like for $1.5°$. Overall, for the same elevation
the correlation length scales increase with the increasing height. This can not be due to the beam broadening effect since it is
omitted in both model runs. The reason for this can be attributed to the fact that for higher heights the radar beams have to
penetrate longer distances (see Fig. 2b), suffering more from attenuation and likely the other errors, which may cause longer
correlation length scales. Moreover, for the same height, the correlation length scales exhibit slight sensitivity to different
185    elevations.

The estimated standard deviations of the RE for radial wind are depicted in Figs. 7a. The biases of the RE are approximately
zero (not shown). The numbers of samples used vary from $10^4 \sim 10^7$ at the surface to $\sim 10^4$ at the top (see Fig. A1). Standard
deviations are similar at lower levels for all elevations and they increase with height while standard deviations for lower
elevations increase faster. The increase with the height can be simply due to increasing radial wind speed with height (see
190    Fig. 7b). Furthermore, since the lower elevations are more sensitive to the horizontal wind, larger standard deviations for the
lower elevations suggest that the RE of the horizontal component of the radial wind may dominate the error. The reason for

this can be that with the increasing height anvil regions are approached where divergent convective outflows occur and winds move in different horizontal directions and slight spatial shifts of cells in simulations can lead to large errors.

The horizontal correlations of the RE for the radial wind are given in Fig. 8. The numbers of samples used for each separate distance and for each elevation are about $10^6 \sim 10^8$ (see Fig. A3). For the elevation $0.5°$, the correlation length scales at heights of 0.2, 1, 2, 3 and 4 km are estimated, which are about 5, 10, 6, 6 and 6 km. For $1.5°$, the correlation length scales at all heights are estimated, which are about 4, 7.5, 6, 5.5, 5.5 and 6 km. For $3.5°$, the correlation length scales at all heights are about 2.5, 5.5, 5.5, 5.5, 5.5 and 6 km. For $5.5°$, the correlation length scales are about 2, 4, 5, 5.5, 5.5 and 6 km. Therefore, except at 1 km for $0.5°$ and $1.5°$ (the reason for this is unidentified), the length scales lengthen with the increasing heights for the same elevation. Furthermore, the length scales at the same height become shorter for higher elevations, especially for lower heights, which can be due to the fact that radial wind is a directional measurement and for the same height the lower elevations see more horizontal components of the radial wind than vertical component and the error correlation of the horizontal wind has a much longer length scale (see Figure 5 of Zeng et al. 2019).

## 4.2 Observation error statistics estimated by the Desroziers method

### 4.2.1 Experimental settings

To apply the Desroziers method, data assimilation experiments are run with hourly update from 03 June to 17 June 2019. During this summer time, severe thunderstorms affected large parts of Germany. We use the operational ICON-LAM KENDA system. The ensemble size is 40 and a deterministic run is updated by the Kalman gain for the ensemble mean. The main data assimilation scheme is the LETKF, via which conventional observations including radiosondes (TEMP), wind profilers (PROF), aircraft reports (AIREP) and synoptic surface observations (SYNOP) as well as MODES data (Lange and Janjić, 2016) are assimilated. More details about the usage of those observations can be found in Schraff et al. (2016).

In addition, radar observations are also directly assimilated by the LETKF via using the radar observation operator EMVO-RADO. Prior to assimilation, signal processor filters (Werner, 2014) are applied to radar observations for quality control. Due to high density, radar observations are thinned in time and space. Temporally, only the latest 5 min radar observations prior to the analysis time are assimilated (Zeng et al., 2018, 2019, 2020, 2021a, b; Gastaldo et al., 2021). Spatially, the superobbing technique is applied to each PPI scan (see Fig. 3), with the goal of obtaining homogeneous distributions of observations in horizontal. The following explains how it works: first, a horizontal Cartesian grid with a desired resolution $d_o$ is defined. Second, for each Cartesian grid point, the algorithm searches for the closest radar bin. If the bin is not too close to the radar station (within radial range of 10 km), it will be regarded as the center of superobbing. Third, a wegde-shaped area around the center bin is defined by the radial range interval and the azimuth width. The radial range interval is given by $\pm\frac{d_o\sqrt{2}}{2}$ and the azimuth width at the radial range $r_0$ is given by $\pm\arctan\left[(d_o\sqrt{2}/2)/r_0\right]$. The superobservation is then created by averaging raw observations within the wedge area. Note that: 1) before superobbing, the modelled radial wind is dealiased (the folding speed is 32 m/s), and all reflectivities lower than 0 dBZ are set to be 0 dBZ in both simulations and observations and treated as clear-air reflectivity data as mentioned above. 2) If there are fewer than three raw observations within the wedge area, the super-

observation will be discarded. Additionally for the radial wind, the azimuth width is limited to $20°$ (an illustration of variations

of the azimuth width with the height is given in Fig. A4), and if the standard deviation of raw observations exceeds 10 m/s,

the superobservation will be discarded. Second and third steps are repeated for all PPI scans. As noted by Waller et al. (2019),

this superobbing technique may create error correlations since the same raw observations may be accounted for in neighboring

superobservations. In this study, we set $d_o = 5$ km. Examples of superobservations at elevation $0.5°$ are given in Fig. A5. In

addition to the LETKF, the latent heat nudging (LHN) is performed for each ensemble member and the deterministic run to

assimilate radar-derived precipitation rates (Stephan et al., 2008).

    For specification of the $\mathbf{R}$, a diagonal matrix is used. For the radar reflectivity, $\mathbf{R} = 10 \cdot 10 \cdot \mathbf{I}$ [dBZ$^2$], where $\mathbf{I}$ is the identity

matrix. For the radial wind, $\mathbf{R} = 2.5 \cdot 2.5 \cdot \gamma \cdot \mathbf{I}$ [m$^2$/s$^2$]. Since the IE of radial wind measurements are usually large if onsite

reflectivities are too small, a scaling factor $\gamma$ is introduced, which varies between 1.0 and 10.0 depending on reflectivities and is

determined by the EMVORADO. For reflectivities smaller than 0 dBZ, $\gamma = 10.0$, for reflectivities greater than 10 dBZ, $\gamma = 1.0$,

and for reflectivities inbetween, $\gamma$ decreases linearly. Before (super)observations are assimilated, a first guess check is carried

out (i.e., the innovation of the deterministic run must be smaller than three times the standard deviation of the innovation).

For localization, an adaptive localization is applied in horizontal for conventional data, whose radius is bounded between 50

and 100 km, and the radius of vertical localization varies with altitude from 0.0075 to 0.5 in logarithm of pressure. For radar

observations, the localization radius in horizontal is constant and set to be 16 km. The observations are weighted by the 5-

th order Gaspari-Cohn function (Gaspari and Cohn, 1999). For inflation, adaptive multiplicative inflation, relaxation to prior

perturbations (RTPP, Zhang et al. 2004) and large-scale additive noise (Zeng et al., 2018) are jointly applied. The prognostic

variables are updated at the analysis step except the precipitating variables (i.e., $q_s$, $q_g$ and $q_r$). Also note that boundary layers

including 67 km lateral boundaries at each side and heights above 300 hPa (approximately 10 km) are not updated.

### 4.2.2    Results

    As Waller et al. (2019), we use the first guess and analysis departures of the deterministic run to calculate the statistics of the

OE. Compared to statistics shown in Section 4.1, the OE obtained here are subject to new error sources such as the IE and

FE, as well as a larger RE since the real radar observations have a even finer resolution than 1 km that is used in Section 4.1.

Fig. 9a shows vertical profiles of the estimated standard deviations of the OE for reflectivity data $\geq 0$ dBZ at elevations $0.5°$,

$1.5°$, $3.5°$ and $5.5°$. The numbers of samples used vary between $10^3 \sim 10^6$, and the higher the elevations is, the fewer samples

are available (see Fig. A6). It is noticed that the standard deviations vary between 4 and 7 dBZ, which is much smaller than 10

dBZ that is used to assign $\mathbf{R}$. This is due to the treatment setting all negative values of reflectivity data equal to 0 dBZ in both

observations and simulations for assimilating clear-air reflectivity data, which reduces first guess departures and thus results in

smaller estimated errors (see also Zeng et al. 2021a). To mitigate this problem, only reflectivity data $\geq 5$ dBZ are evaluated in

Fig. 9d, in which standard deviations of all elevations (except $0.5°$ that is strongly contaminated by ground clutters) increase

till about 3 km and then decrease till about 5 km before increasing till around 7 km and decreasing again to the top. Moreover,

the standard deviations of different elevations are quite different, and at the same height the lower elevations are, the larger the

standard deviations are. This can be attributed to a larger IE that occurs while the lower elevations scanning longer distances

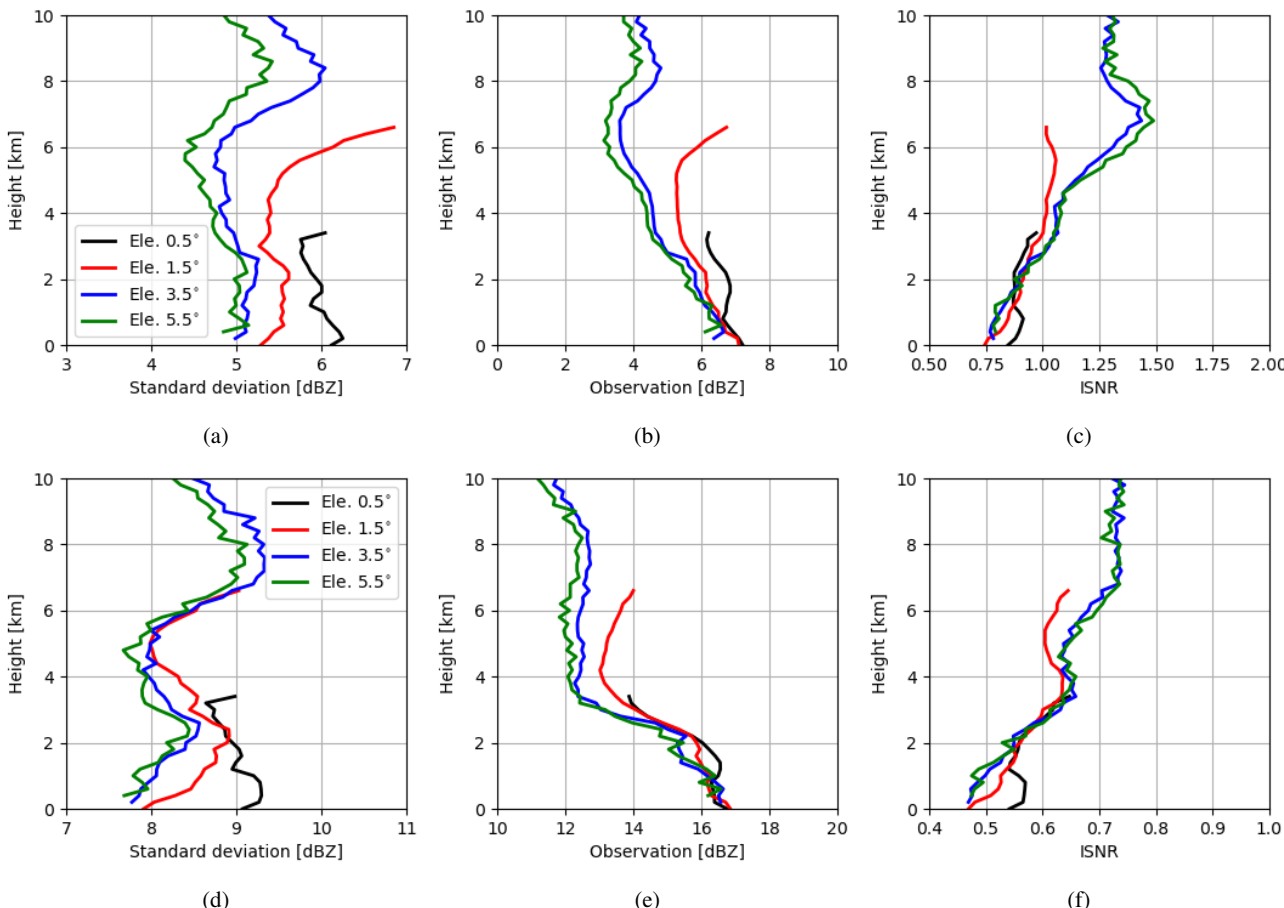

**Figure 9.** a) Vertical profiles of the estimated standard deviations of the OE for reflectivity data $\geq 0$ dBZ at elevations $0.5°$, $1.5°$, $3.5°$ and $5.5°$ b) Vertical profiles of observed reflectivity data $\geq 0$ dBZ, averaged over all samples; c) ISNR, i.e, standard deviations in a) are divided by observation data in b) for each elevation, respectively; (d-f) for reflectivity data $\geq 5$ dBZ. For each elevation, the samples are binned every 200 m in the vertical. Since model states are not updated above 10 km, standard deviations continuously increase (see Waller et al. 2019). Here we show results only up to 10 km.

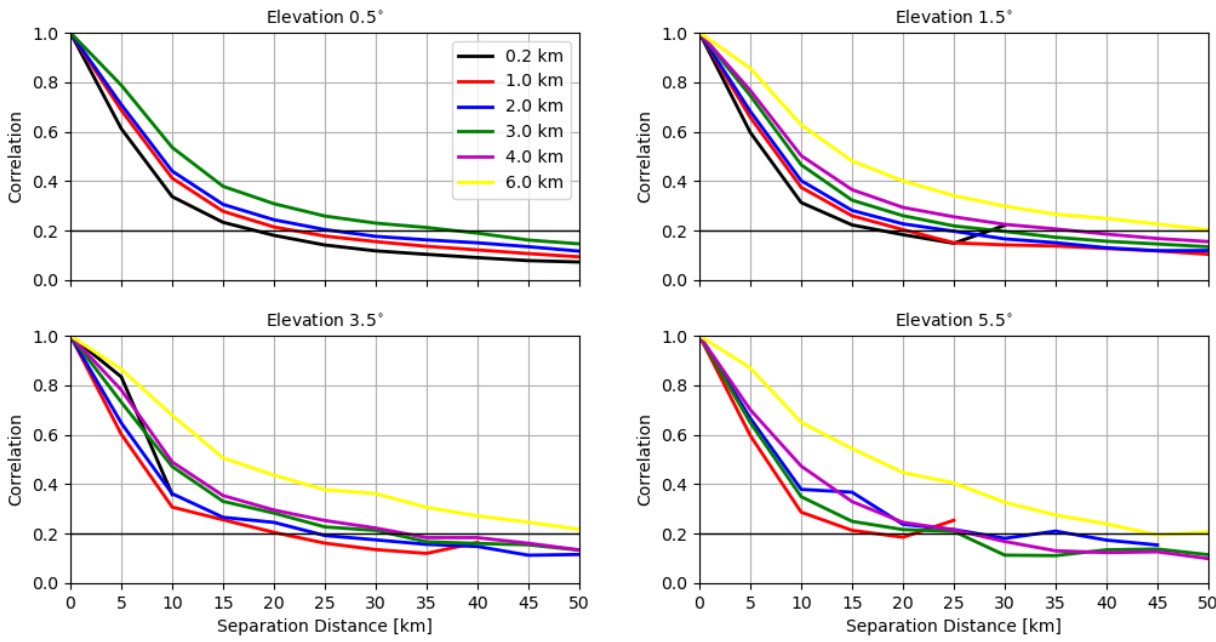

**Figure 10.** The estimated horizontal correlations for the OE of reflectivity data $\geq 5$ dBZ for elevations $0.5°$, $1.5°$, $3.5°$ and $5.5°$ at heights of 0.2, 1.0, 2.0, 3.0, 4.0 and 6.0 km. For each elevation, the samples are binned every 5 km in the separation distance.

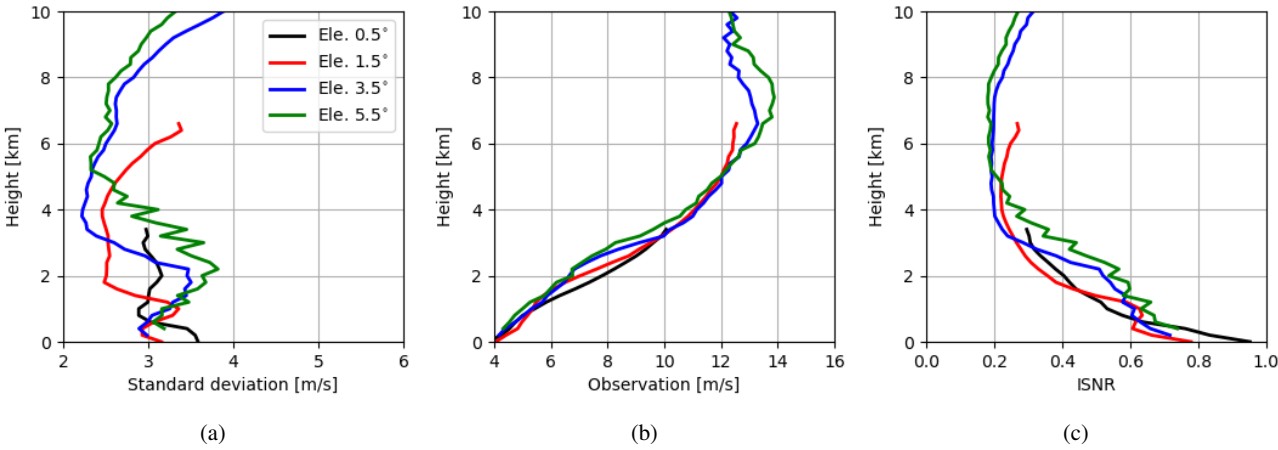

(a)        (b)        (c)

**Figure 11.** The same as Fig. 9 but for radial wind data

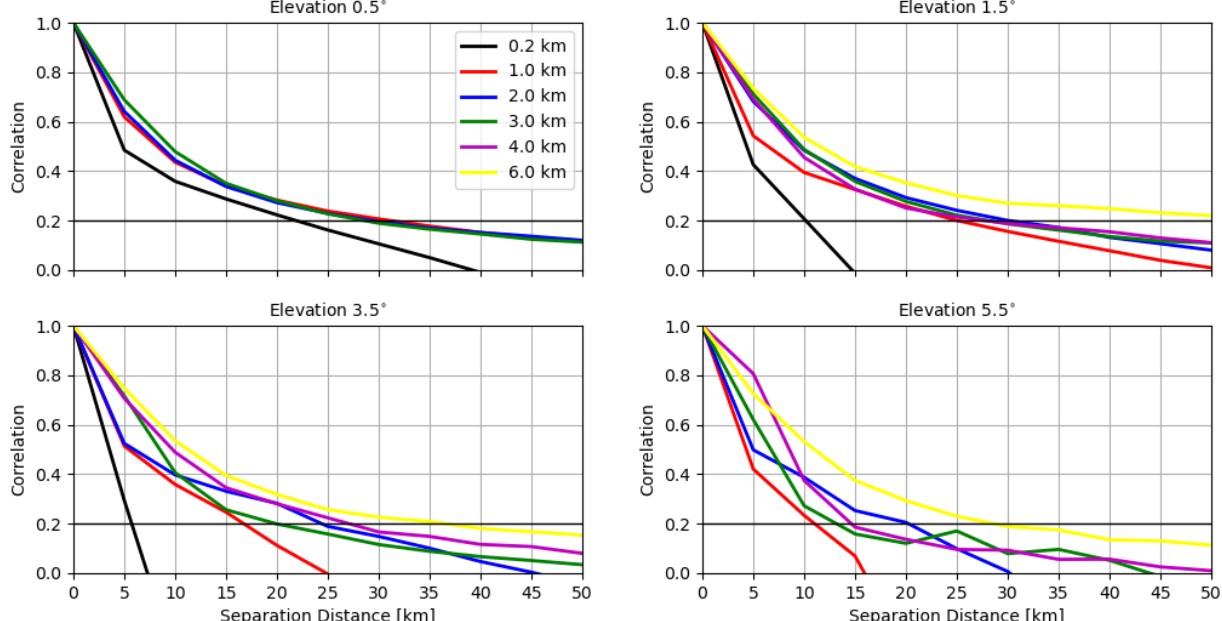

**Figure 12.** The same as Fig. 10 but for radial wind data

as well as to a larger FE that the pulse volumes of lower elevations are larger (see Fig. 3) and the beam broadening effect is
omitted in the EMVORADO though. Finally, both Figs. 9c and 9f show that the ISNRs of all elevations increase till around 7
km and then either slightly decrease or halt.

Fig. 10 shows the horizontal correlations of the OE for reflectivity data $\geq 5$ dBZ. (correlations for reflectivity data $\geq 0$ dBZ
are not shown because they exhibit very similar patterns as in Fig. 10). The numbers of samples used vary between $10^3 \sim 10^6$
(see Fig. A7). For the elevation $0.5°$, the correlation length scales at the the height of 0.2, 1.0, 2.0 and 3.0 km are given, which
are about 20, 25, 25 and 40 km. For $1.5°$, the correlation length scales at the the height of 0.2, 1.0, 2.0, 3.0, 4.0 and 6.0 km are
given, which are about 20, 20, 25, 30, 35 and 50 km. For $3.5°$, the correlation length scales at the the height of 1.0, 2.0, 3.0,
4.0 and 6.0 km are given, which are about 20, 25, 30, 35 and 50 km. For $5.5°$, the patterns of correlations are comparable to
$3.5°$ with some fluctuations due to small numbers of samples used. To sum up, for the same elevation the correlation length
scales increase with the height and that for the same height the correlation length scales exhibit little sensitivity to different
elevations.

Fig. 11a shows vertical profiles of the estimated standard deviations of the OE for radial wind at elevations $0.5°$, $1.5°$,
$3.5°$ and $5.5°$ and the numbers of samples used are given in Fig. A6. The variation patterns of the standard deviations are
similar to Waller et al. (2019) although the values here are greater due to the application of the scaling factor for $\mathbf{R}$ of radial
wind, which inflates $\mathbf{R}$ where small reflectivities (i.e., between 0 and 10 dBZ) are observed. The standard deviations at all
elevations except $0.5°$ reach their maxima at different height between 1 and 6 km, and the higher the elevation is, the larger
the maximum is. Since higher elevations see more in the vertical, this suggests that the vertical component of the radial wind

is not well reproduced. On one hand, it can be related to the misrepresentation of $w$ in the model (see large differences caused by changing the model resolution from 2.1 to 1 km in Fig. 5 and considering that for heights till 6 km the radar beamwidth is much shorter than 1 km for higher elevations, see Fig. 2a). Small-scale strong updrafts often occur within these heights, which radar observations can much better resolve than the model, leading to a larger RE. On the other hand, it can be also related to the misrepresentation of the terminal fall speed of hydrometeors, which can be due to the parametrization in the microphysical scheme and which can also be due to neglecting reflectivity weighting of the radial wind in the EMVORADO. It is shown in Zeng et al. (2016) for an idealized supercell case that the hydrometeor terminal fall speed with the reflectivity weighting can be maximal 8 m/s faster than without. This causes a larger FE. With increasing height, the misrepresentation of the vertical component becomes less (due to coarser resolution of observations with the increasing pulse volume) or less pronounced (due to the increasing horizontal wind speed), and the standard deviations of the OE become smaller. Above 6 km, the standard deviations start to increase with the height and the increase for the lower elevations is faster, which is because errors in the horizontal component of the radial wind are dominating as anvil regions are approached. In Fig. 11c, it can be better seen that the ISNR decreases with the height up to 6 km due to better representation of the vertical component and then increases when it gets closer to anvil regions.

Fig. 12 shows the horizontal correlations of the OE for the radial wind and the numbers of samples used are given in Fig. A8. For the elevation $0.5°$, the correlation length scales at the the height of 0.2, 1.0, 2.0 and 3.0 km are given, which are about 20, 30, 30 and 30 km. For $1.5°$, the correlation length scales at the the height of 0.2, 1.0, 2.0, 3.0, 4.0 and 6.0 km are given, which are about 10, 25, 30, 30, 30 and longer than 50 km. For $3.5°$, the correlation length scales at the the height of 1.0, 2.0, 3.0, 4.0 and 6.0 km are given, which are about 5, 15, 25, 20, 30 and 35 km. For $5.5°$, the correlation length scales are generally a bit short than for $3.5°$. Overall, it still can be said that for the same elevation the correlation length scales increase with the height and the lower elevations generally exhibit longer correlation length scales for the same height.

### 4.3 Discussion

In this section, we discuss the differences between the RE and OE for both reflectivity data $\geq 5$ dBZ and radial wind data.

For reflectivity data $\geq 5$ dBZ, it is noticed in Fig. 13a that the variations of standard deviations of the OE follow those of the RE till about 7 km, indicating that the RE may be a dominant error source at these heights. Above 7 km, the standard deviations of the RE are overestimated since the model produces too high reflectivities (cf. Figs. 4e and 9e). This can be caused by the constant value of the slope intercept parameter $N_0$ in the particle size distribution function of the one-moment microphysical scheme for the graupel, which may be too small for anvil clouds of the deep convection, leading too much large graupel over there. For both RE and OE, it holds that for the same elevation the correlation length scales increase with the height (see Fig. 13c) and that for the same height the correlation length scales exhibit no sensitivity to different elevations (see also Fig. 13e). However, the correlation length scales of the OE are much longer than those of the RE. As argued in Waller et al. (2016c), the increasing pulse volume at longer distances could contribute to this since the the beam broadening effect is omitted in the EMVORADO. In addition, the superobbing can be also responsible for longer length scales. In an experiment

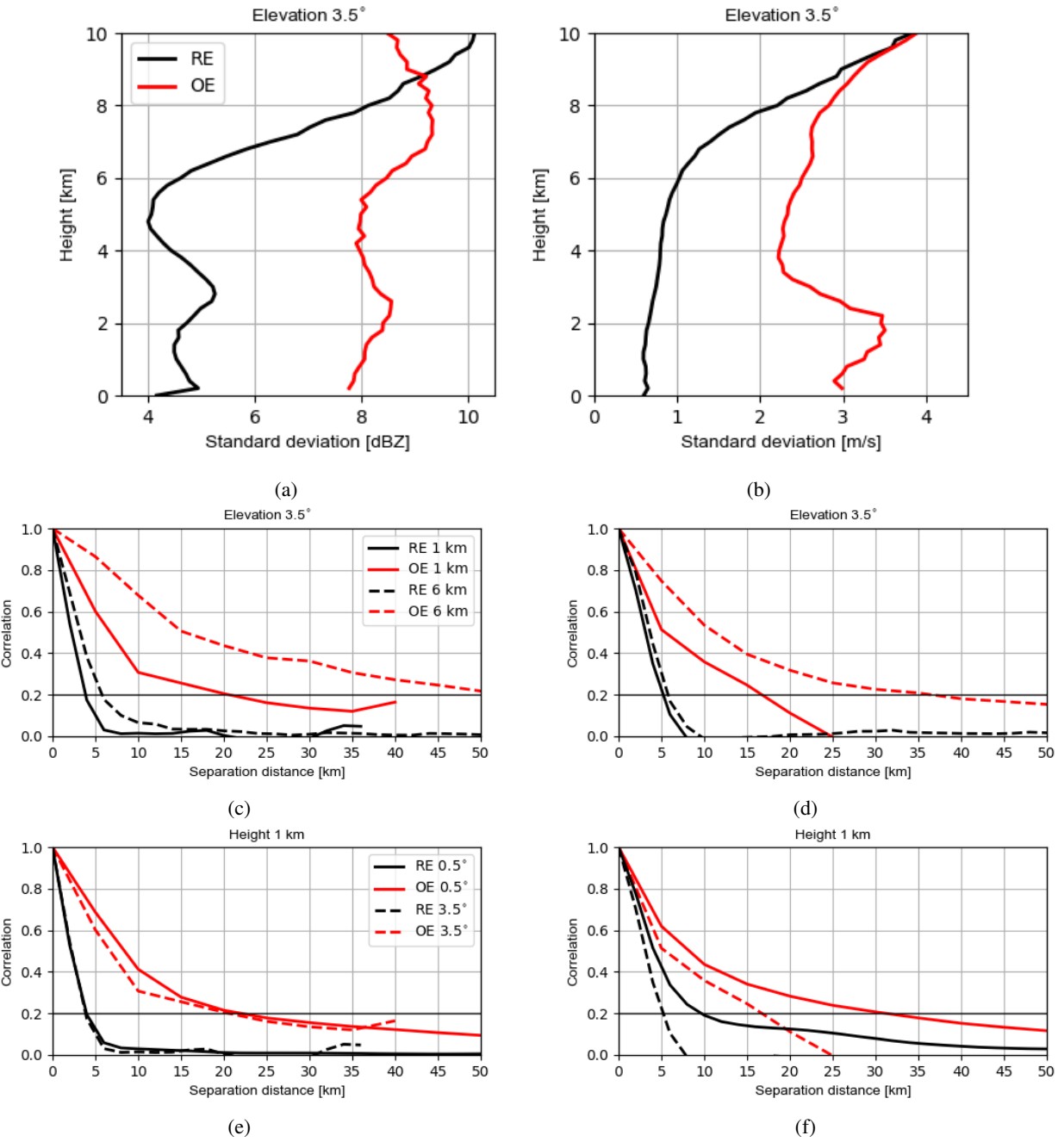

**Figure 13.** a) Vertical profiles of the estimated standard deviations of the RE and OE at elevation $3.5°$ for reflectivity data $\geq 5$ dBZ and b) radial wind; c) The estimated horizontal correlations for the RE and OE at elevation $3.5°$ at heights of 1 and 6 km for reflectivity data $\geq 5$ dBZ and d) for the radial wind; e) The estimated horizontal correlations for the RE and OE at the height of 1 km for elevations $0.5°$ and $3.5°$ for reflectivity data $\geq 5$ dBZ and f) for the radial wind

with the superobbing resolution of 10 km (here 5 km), the lengths scale are generally increased by 5 km (not shown). The error caused by the superobbing is usually considered as a type of the FE (Janjic et al., 2018).

For radial wind, the variations of standard deviations of the RE and OE also share some similarities except at the lower heights as shown in Fig. 13b. The differences are caused mainly by the FE arising from the microphysical scheme and the observation operator as mentioned before. The correlation length scales of the OE are also much longer than those of the RE

and for the same elevation the correlation length scales of the RE and OE increase with the height. These are the same as for reflectivity data. Some differences are also noticed. For instance, the correlation length scales of the RE and OE for radial wind are less sensitive to the heights (cf. Figs. 13c and 13d) and more sensitive to the elevations (cf. Figs. 13e and 13f). The former one can be explained by that radial wind data are less affected by attenuation, and the latter one is because the radial wind is a directional measurement. Generally, one could expect that the correlation length scales for radial wind are longer than for

reflectivity at lower elevations since the error correlation length scales of the horizontal wind are longer than those of mixing ratio variables (Zeng et al., 2019), but for higher elevations, the role of vertical component of the radial wind amplifies and shortens the correlation length scales.

## 5 Summary and outlook

An adequate specification of the observation error (OE) covariance can be beneficial for convective-scale radar data assimilation

since the radar measurements are dense and their errors are strongly correlated. The Desroziers method has been used in previous studies to calculate the variances and correlations of the OE for radial wind data (e.g., Waller et al. 2016c, 2019). However, the estimated statistics are not fully understood since they are composed of contributions from different sources such as instrument error (IE), the observation operator error and pre-processing or quality control error as forward model error (FE) and the error due to unresolved scales and processes (RE) and it is difficult to distinguish individual portion (Janjic et al., 2018).

To better understand possible contributions of the RE, another approach is proposed, which assumes a high resolution model run as truth and a low resolution model run as a truncation, and a set of samples for truncation error are created and used to approximate the statistics of the RE. It is noted that the standard deviations for reflectivity data are usually underestimated by the Desroziers method due to setting all negative reflectivity values to 0 dBZ (treated as clear-air reflectivity data). To mitigate this problem, we focus on the statistics of RE and OE for reflectivity data $\geq 5$ dBZ in this work.

We have run the ICON-D2 model (equipped with the radar observation operator EMVORADO) with resolutions of 1.0 and 2.1 km in a coupled manner for a summer convection period. A sufficient amount of samples for truncation error in radar observation space have been obtained. The statistics of samples are evaluated for each elevation. It is found for the reflectivity data $\geq 5$ dBZ that the standard deviations of the RE are similar for all elevations and are approximately proportional to the true signals. For the same elevation, the horizontal correlation length scales increase with height, which can be attributed to the fact

that for higher heights the radar beams have to travel through longer distances. For the same height, the correlation length scales are not sensitive to different elevations. Similar can be seen for the radial wind except that at the same height the length scales become shorter for higher elevations, especially for lower heights, which can be due to the fact the radial wind measurement has

a directional component and for the same height the lower elevations see more horizontal components of the radial wind than vertical component and the error correlation of the horizontal wind has a much longer length scale. Furthermore, the correlation

length scales for reflectivity are generally a bit shorter for lower elevations, this may be because the error correlation length scales of mixing ratio variables are shorter than those of the horizontal wind, but for higher elevations, the vertical component of the radial wind shortens the correlation length scales.

We have also performed data assimilation experiments, using the ICON-LAM KENDA system with operational settings for another summer convection period. We have used the Desroziers method to estimate the statistics of the OE. Results show that

the standard deviations of reflectivity $\geq 5$ dBZ exhibit a comparable pattern to that of the RE, indicating the RE is a dominant error source. The difference to the RE also exists. The standard deviations of different elevations are quite different, i.e., at the same height, the lower the elevations are, the larger the standard deviations are. This can be caused by a larger IE that occurs while the lower elevations scanning longer distances and a larger FE that the pulse volume of lower elevations are larger but the beam broadening effect is omitted in the EMVORADO. Furthermore, it is seen that representation error is considerably

overestimated above about 7 km. This is likely attributed to the deficiency in the microphysical scheme, leading too much large graupel at these heights. For radial wind data, the standard deviations of the OE are much larger than those of the RE, this is mainly due to the the application of the scaling factor for the OE used in the data assimilation system. The scaling factor inflates the OE where the reflectivity values are too small ($\leq 10$ dBZ). Besides, the standard deviations are especially large between heights of 1 and 6 km for the higher elevations. This can be caused by the misrepresentation of the vertical wind in

the model since the the vertical wind is very sensitive to the model resolution, it can also be caused by the misrepresentation of the terminal fall speed of hydrometeors due to inaccuracy in parametrization of the microphysical scheme and to neglection of reflectivity weighting of the radial wind in the EMVORADO. With respect to the correlation length scales of the OE, for both reflectivity and radial wind data, the length scales behavior similarly as those of the RE but the length scales are much longer. Among the possible error sources mentioned above, the application of the superobbing is another reason. Shorter superobbing

resolution reduces the length scales. Overall, it is successful to use the statistics of truncation error in observation space to better understand the statistics of the OE estimated by the Desroziers method for both reflectivity and radial wind data. The RE contributes greatly to the variances and defines several features in the correlation length scales.

It is noted that using the Desroziers method for all reflectivity data including clear-air reflectivity data always results in too small error variances. Since the Desroziers method tends to produce too small variances by its nature as shown in a

number of studies (Weston et al., 2014; Bormann et al., 2016), one should greatly inflate them if one considers using them for assignment of the OE for reflectivity in data assimilation. However, since reflectivity data and clear-air reflectivity data are associated with different error characteristics as for example the all-sky radiances (error standard deviations in clear sky are much smaller than in heavy cloud or precipitation, Geer and Bauer 2011; Chambon et al. 2014), one could consider treating them as different data during data assimilation. In contrast, the variances of the radial wind is considerably large due

to the scaling, it may be unnecessary to inflate them for the further use. Comparing the statistics of the RE and OE, we see the potential in improving the microphysical scheme and the necessity of using a more comprehensive configuration of the EMVORADO (or even improvement). Moreover, results presented here are based on the convective period in the summertime.

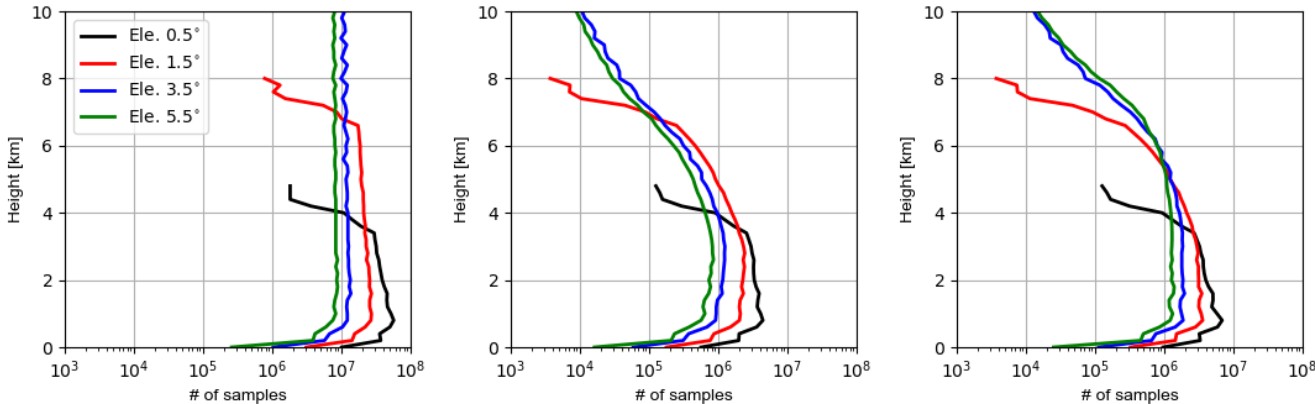

**Figure A1.** Vertical profiles of numbers of samples used for estimating standard deviations of the RE for reflectivity data $\geq$ (left) and $\geq 5$ dBZ (middle) and for radial wind (right) at elevations $0.5°$, $1.5°$, $3.5°$ and $5.5°$

The applicability of those results to other periods such as wintertime cyclonic systems needs further investigation. However, some studies have been done by the other centers. For instance for the Met Office UKV model with the 3D-VAR scheme,
the estimated OE statistics (based on Desroziers method) for radial wind are qualitatively similar to those in the summertime (Waller et al., 2016c), and for reflectivity, Kouroupaki (2019) shows that the estimated standard deviations of the OE in Winter are larger than in Summer and that they increase with reflectivity values. Since the current superobbing method can lead to extra correlations, we may also consider refining this method. Finally, the statistics presented here show the necessity of including correlations in the OE covariance matrix for radar data assimilation and this work can be used as a guideline for selecting
which observations are assimilated and for assignment of the OE covariance matrix that can be diagonal or full and correlated.

## Appendix A: Additional figures

In this section, we provide figures for the numbers of samples used for estimating the statistics of the RE (Figs. A1-A3) and OE (Figs. A6-A8) as well as figures for the superobbing technique (Figs. A4-A5).

*Code and data availability.*

All the data upon which this research is based are available through personal communication with the authors. Access to the source code of the ICON model is restricted to ICON licenses. A license can be obtained for research if following the procedure described at https://code.mpimet.mpg.de/projects/icon-license.

*Author contributions.*

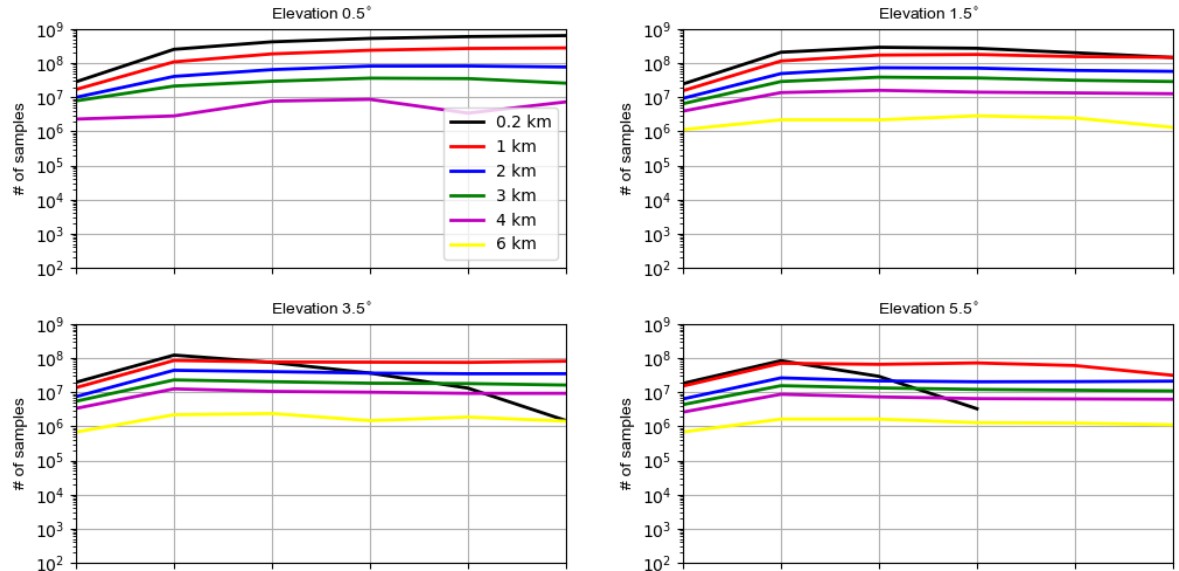

**Figure A2.** The numbers of samples used for estimating horizontal correlations for the RE of reflectivity data $\geq 5$ dBZ for elevations $0.5°$, $1.5°, 3.5°$ and $5.5°$ at heights of 0.2, 1.0, 2.0, 3.0, 4.0 and 6.0 km

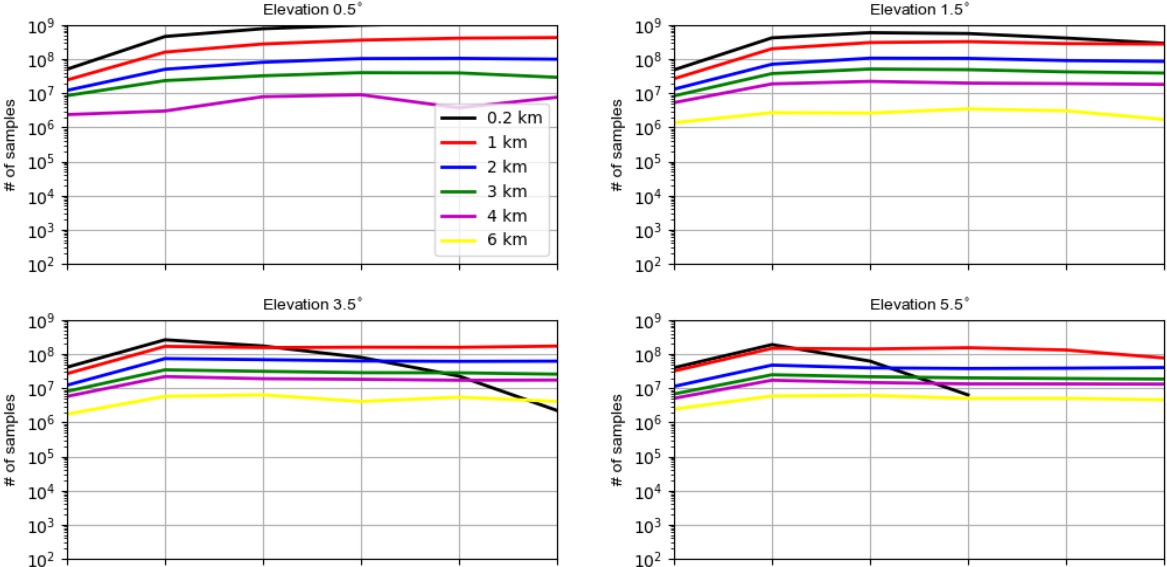

**Figure A3.** The same as Fig. A2 but for radial wind data

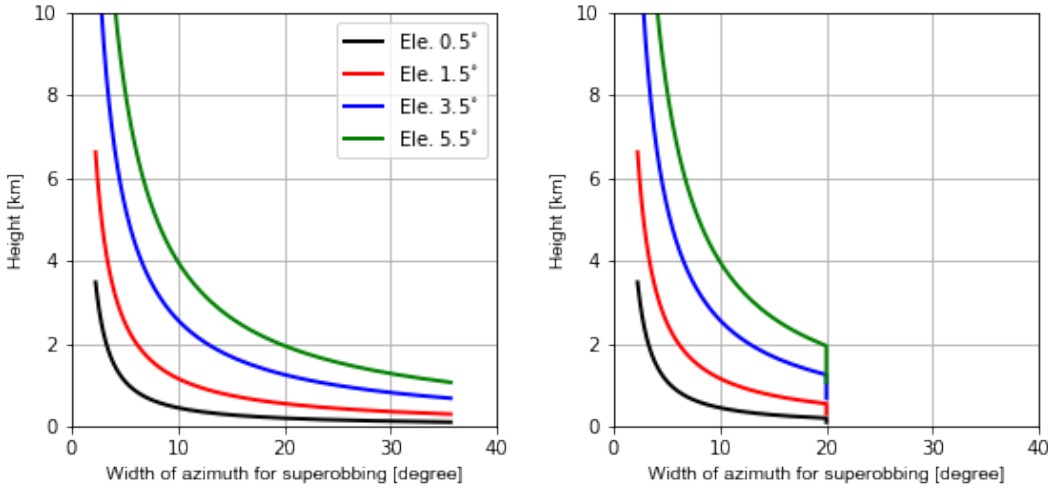

**Figure A4.** Variations of the width of azimuth with height for superobbing of reflectivity (left) and radial wind (right) at elevations $0.5°$, $1.5°$, $3.5°$ and $5.5°$. A maximum of $20°$ is set for superobbing of radial wind.

Y. Zeng and T. Janjic provided the idea, Y. Zeng conducted the data assimilation experiments and wrote the first draft of the
paper. Y. Feng conducted the experiments for creation of samples of truncation error and evaluated them. T. Janjic, U. Blahak and J. Min contributed to the conceptual design of the research project and A. de Lozar, E. Bauernschubert and K. Stephan contributed to discussion and interpretation of the results.

*Competing interests.*

The authors declare that they have no conflict of interest.

*Acknowledgements.* Thanks are given to the DFG (Deutsche Forschungsgemeinschaft) Priority Program 2115:PROM through the project JA 1077/5-1. The work of Y. Feng is sponsored by the China Scholarship Council. T. Janjic is thankful to the DFG for the Heisenberg Award JA 1077/4-1. The work of U. Blahak is supported by the Innovation Programm for applied Research and Development (IAFE) of Deutscher Wetterdienst in the framework of the SINFONY project. Thanks are also given to Roland Potthast from the DWD and Leonhard Scheck from the Hans Ertel Centre for Weather Research (Weissmann et al., 2014; Simmer et al., 2016) at the LMU for technical supports.

*Financial support.*

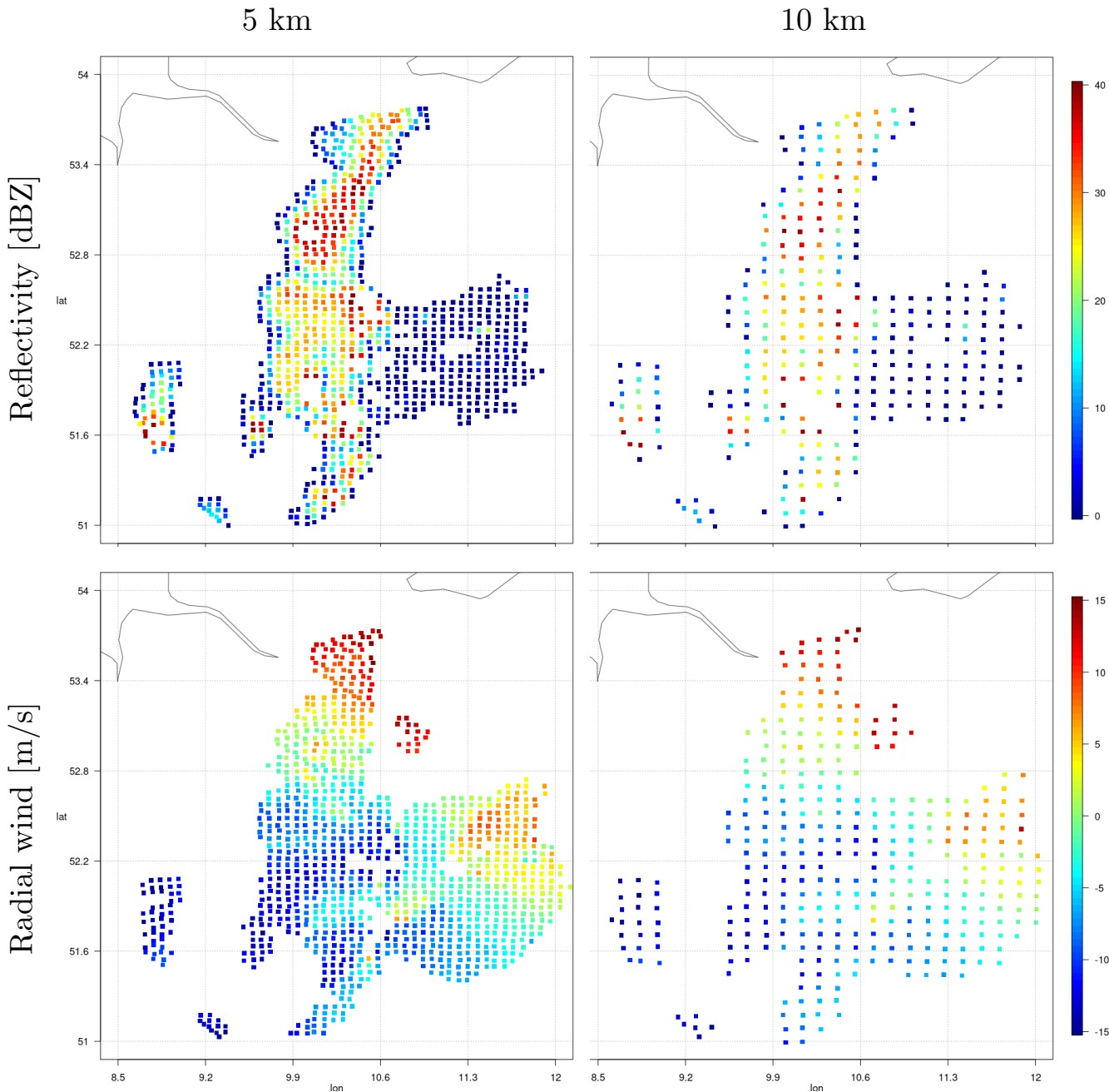

**Figure A5.** Illustration of superobbing for reflectivity data $\geq 0$ dBZ (upper) and radial wind (lower) with resolution $d_o$ = 5 km (left) and $d_o$ = 10 km (right) at elevation $0.5°$ for the radar station 10356 at 13:00 UTC June 3, 2019

This research has been supported by the DWD, the DFG (Deutsche Forschungsgemeinschaft) Priority Program 2115:PROM through the project JA 1077/5-1 and the China Scholarship Council.

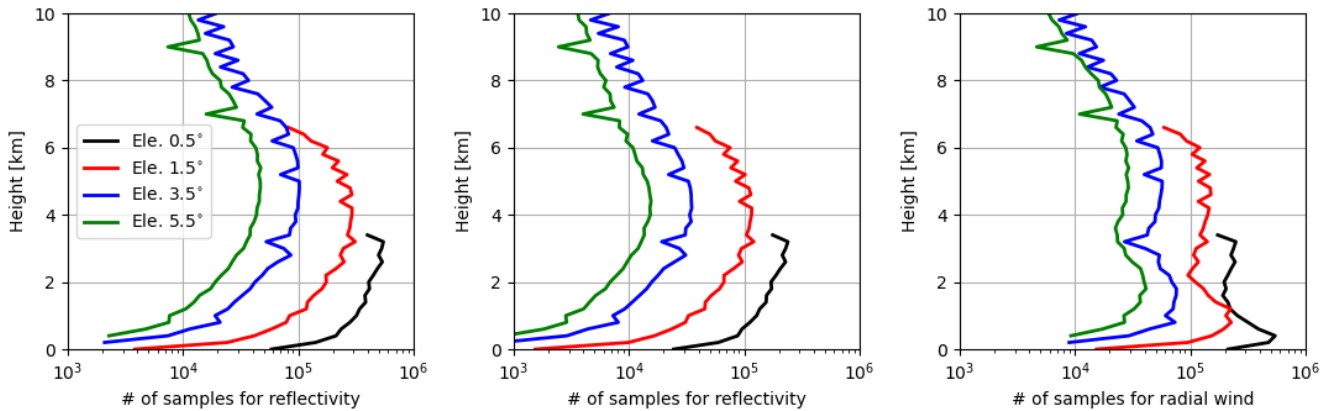

**Figure A6.** Vertical profiles of numbers of samples used for estimating standard deviations of the OE for reflectivity data $\geq 0$ dBZ (left) and $\geq 5$ dBZ (middle) and for radial wind (right) at elevations $0.5°$, $1.5°$, $3.5°$ and $5.5°$

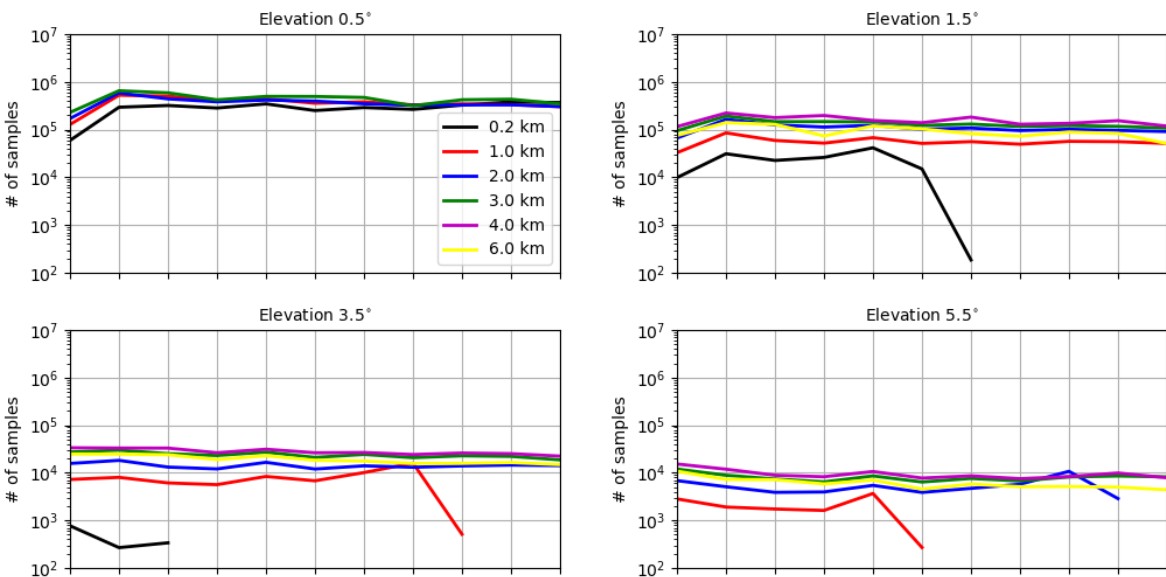

**Figure A7.** The numbers of samples used for estimating horizontal correlations for the OE of reflectivity data $\geq 5$ dBZ for elevations $0.5°$, $1.5°,3.5°$ and $5.5°$ at heights of 0.2, 1.0, 2.0, 3.0, 4.0 and 6.0 km

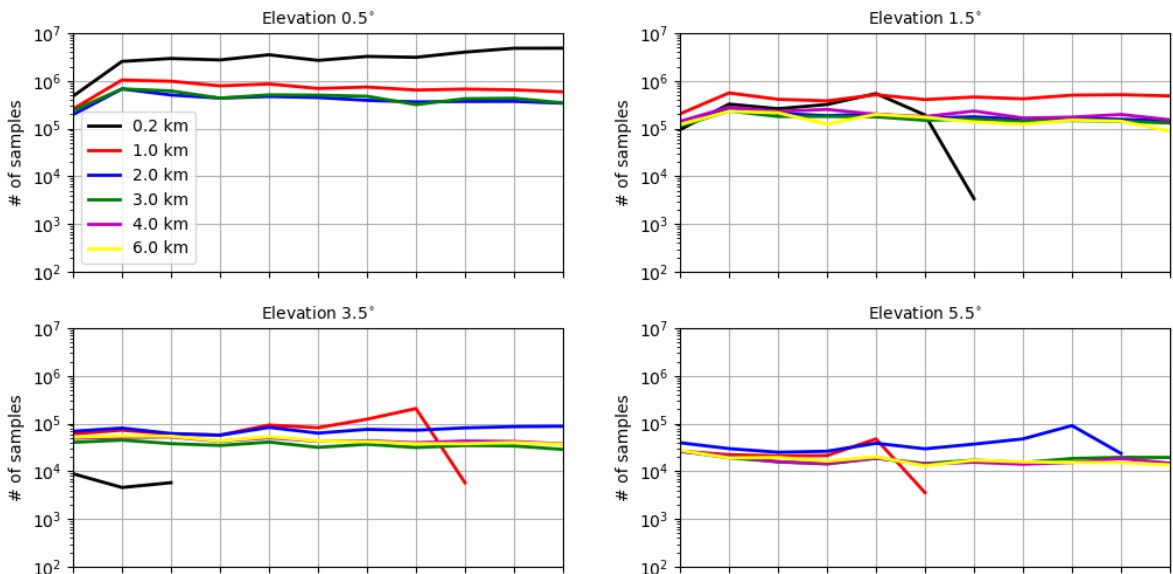

**Figure A8.** The same as Fig. A7 but for radial wind data

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
