# Peer review of "Interpreting estimated Observation Error Statistics of Weather Radar Measurements using the ICON-LAM-KENDA System"

_Atmospheric Measurement Techniques, 2021_

## Author Comment (AC1)

**To Reviewer 1**

This work tries to estimate components of the observation error for radar reflectivity and radial wind measurements using two complementary techniques: the difference between high and low resolution forecasts as an indicator of representation error, and the well-known Desroziers approach that is intended to estimate the total observation error. There are substantial differences between the two estimates, which the paper attempts to explain. This is useful work on the tricky subject of assigning observation errors to precipitation-sensitive observations.

**Answer:** Thank you very much for your kind acknowledgment of our work.

However, there are major issues with the work as currently presented. The first is an important scientific issue: the results are given for a precipitation-affected sample based on a 5 dBZ reflectivity threshold; however little attention has been paid to how that threshold has been defined and how the definition might affect the sample of observations being examined. Given that the threshold definition must be different in the two different techniques, this might be a major cause of variation between the two. Second, there is a lack of high-level figures to help synthesise the results; some of the figures that are presented are given at a perhaps excessive level of detail. Hence, major revisions are recommended.

**Answer:** In the revision of the manuscript we hopefully clarified the reasons behind use of a threshold (see also answer to question 1 below). We also added figures that show results for two threshold values. In addition we added a new section (4.3) that focuses on comparison between the two methods of computing observation error statistics.

**Major issues**

1) Sample creation using the 5 dBZ threshold.

A first issue is that the precise application of the 5 dbZ threshold to identify precipitation is not described in detail. For the representation error calculation, the threshold described on line 128 may be applied to the low-resolution data, the high-resolution data, or both. Hence without further details, this is ambiguous. For the Desroziers calculation, the threshold may be applied to the observations or to the model simulations, or to both; again this is not specified. It is also not fully clear if the threshold is applied per observation location, and how the threshold relates to the ensemble if one is used (is it the ensemble mean, control or ensemble members being used?)

In the literature on all-sky passive microwave assimilation, it is well-recognised that sampling issues need to be treated with care. Given that the location of precipitation in the forecast and the observation may be different, a "precipitation" sample based on the model precipitation alone will exclude many locations where the observations have precipitation but the model does not, and vice-versa. Depending on exactly how the thresholds are applied, very different

bias and standard deviation characteristics may be observed. See for example:

"Observation errors in all-sky data assimilation", Geer and Bauer, 2011, https://doi.org/10.1002/qj.830

"Assessing the impact of pre-GPM microwave precipitation observations in the Goddard WRF ensemble data assimilation system", Chambon et al., 2014, https://doi.org/10.1002/qj.2215

**Answer:** The reflectivity $Z$ has initially units of $[mm^6/m^3]$, however, because numerical values of $Z$ may span several orders of magnitude, it is convenient to use a logarithmic scale in practice, defined as units of dBZ $= 10\log_{10}\left[\frac{Z}{1mm^6/m^3}\right]$. For instance, $10^5\ mm^6/m^3 = 50$ dBZ; $1\ mm^6/m^3 = 0$ dBZ; $10^{-5}\ mm^6/m^3 = -50$ dBZ, and $0\ mm^6/m^3 = -\infty$ dBZ, for which -99.99 dBZ is used to represent $0\ mm^6/m^3$ in the radar forward operator. It is noticed that for very small reflectivities, their differences in units of $mm^6/m^3$ are trivial but significant in units of dBZ. This can be problematic if assimilating those data, which could lead to unrealistically large increments and spurious convection (Zeng et al. 2021a). Therefore, it is very well established in radar data assimilation community to set a threshold value for small reflectivities, in the operational setup of KENDA, it is 0 dBZ, which means all reflectivities values lower than 0 dBZ are set to 0 dBZ, and we call 0 dBZ data as "no reflectivity data". The same threshold value is set to all observations and to all simulated reflectivities in each ensemble member. Second, due to setting the same threshold value to both observations and backgrounds, the innovations are reduced and the observation error variances are underestimated when computing Desroziers diagnostics (Zeng et al. 2021a). To have better statistics that are not affected by the operation of setting no reflectivity data, we calculate Desroziers diagnostics for reflectivities with positive values, for which 5 dBZ is chosen. We made those points more clear in the text. In addition, we added in conclusion section "since reflectivity data and no reflectivity data are associated with different error characteristics as for example the all-sky radiances (error standard deviations in clear sky are much smaller than in heavy cloud or precipitation, Geer and Bauer 2011; Chambon et al. 2014), one could consider treating them as different data during data assimilation".

In the current manuscript, a particularly striking difference is seen between the samples used in the representation error and Desroziers studies (Fig. 4c and 12c) at higher altitudes. Simulated reflectivity reaches 20 dBZ at 10km in the representation error sample but is just 12 dBZ in the observations in the Desroziers sample. This suggests that the sample at high altitudes in the representation error study is dominated by infrequent deep convection, since this is likely the only thing that can generate greater than 5 dBZ reflectivity at that altitude. The big difference to the observational sample could be explained by model error, but it could also just come from a major difference in the composition of the sample being analysed. These aspects need more attention.

**Answer:** High reflectivities in simulations at higher altitudes are probably caused by the inappropriate value for the slope intercept parameter $N_0$ in the

particle size distribution function of the one-moment microphysical scheme. $N_0$ is empirically estimated by the surface measurements, which may be too small for anvil clouds at higher altitudes, leading too much large graupel over there and overestimated representation error. We added this into the text.

2) Need for higher-level figures to summarise the results.

Figures 6, 8, 13 and 15 give a possibly excessive level of detail. The text has to do a lot of description and further analysis of these figures. It presents many lists of numbers derived from these plots, such as the correlation length scales (see e.g. lines 159 - 164). The results derived from these figures would be better presented and analysed on higher level figures, ideally comparing the representation error study with the Desroziers study on the same figures. This would reduce the need to present long lists of numbers in the text.

Similarly, the most interesting aspects of the study are the comparisons between figures 4 and 12, and Fig 7. versus Fig. 14. It is somewhat inconvenient to have to compare these figures manually. A summary figure combining some of the lines from both could be useful.

**Answer:** In the revision of the manuscript, we moved some of the previous figure panels to Appendix, and added a new section that compares the methods. To this end, we also added Fig. 13 as the summary figure.

**Minor issues**

1) Line 22: the acronym ICON-D2 is not explained, nor its significance (presumably the first application of radar reflectivity assimilation in this framework?)

**Answer:** We rephrased as "The ICON-LAM (ICON-Limited Area Model) is the limited area version of the ICON model and is to replace the COSMO model in the operational forecasting system. The ICON-D2 (D: Deutshcland (Germany); 2: 2 km) is an ICON-LAM setup at approximately 2 km grid spacing, which is restricted to Germany and the neighboring countries and became operational for very short-range forecasting since February 2021". We also emphasized the significance of this study in the text.

2) Line 65: The $\mathcal{H}$ operators in equation 1 need some more explanation. Clearly they are not identical since the model inputs are on different grids. Any interpolation or coarse-graining within these operators needs to be explained. This is particularly important since the observation operator is nonlinear. Hence, the application of $\mathcal{H}$ to the mean of the model forecast may be strongly different to the mean of $\mathcal{H}$ applied to individual locations in the forecast. This maps onto the well-known beam filling effect

**Answer:** Reflectivities are first calculated on the model grid points and then interpolated onto radar coordinates. For radial wind, three wind components are interpolated onto radar coordinates and then radial winds are calculated.

3) Line 73: Although it's partly explained later on, it would be useful to have some words on how the model states $x_a$ and $x_b$ used in equation 2 relate to the model states defined in equation 1, given broadly as $M(x^T)$. Even better, homogenising the notation between these equations 1 and 2 would help define the precise differences in methodology between the two halves of this work (such as highlighting the resolution and / or model differences involved, and/or differences in the observation operators being used).

**Answer:** We added "In the following, we estimate statistics of the RE by using the method from Section 2.1 and statistics of the OE by applying the Desroziers method to an data assimilation experiment with a low resolution model, i.e., $\mathbf{d}_{o-b} = \mathbf{y^o} - \mathcal{H}(\mathbf{x}_b^L)$ and $\mathbf{d}_{o-a} = \mathbf{y^o} - \mathcal{H}(\mathbf{x}_a^L)$".

4) Line 76-77: "$R_{est}$ is optimal in case of ..." - are there any further references to back this up or is it from Desroziers (2005)?

**Answer:** we added Reichle et al. 2002.

5) Figure 3, legend: "Scratch of..." is odd terminology - change the term or explain it.

**Answer:** We changed "Scratch" to "Illustration".

6) All figures in the manuscript, but particularly Figs. 3 and 4: the point markers (such as a square or a circle) are so frequently sampled that they change the width of the lines, making them very thick in places and making it hard to do detailed comparisons between the lines. Consider showing all these graphs with only lines, not lines and symbols.

**Answer:** Done.

7) Figures 2 and 3 are not linked to the text where they appear. In any case there needs to be some early description of the processes of superobbing, and the details of the observation operator, in section 3. Instead these details appear partially, and too late, in section 4.2.1, for example.

**Answer:** We switched the order of Figures 2 and 3. We added more details about the observation operator in section 3. Since the processes of superobbing is only done for data experiments for OE (not for RE), it is appropriately positioned in section 4.2.1.

8) Furthermore, there are some other introductory details missing on the model setups: for example for the models used in the representation error study, it is not clear whether data assimilation is applied to keep the forecast on track, whether (and where from) there are boundary conditions being applied to achieve the same result. An introduction to the models being used in both halves of the study, their similarities and differences, would be very useful around section 3.

**Answer:** For representation error, we are interested in its climatology instead of exact position and intensity of convection, therefore, no data assimilation is applied to the models used in the representation error study. But in both studies, models are driven by hourly boundary conditions.

9) Line 107: "standard deviation and horizontal correlation" ... of what?

**Answer:** We changed it to "standard deviation and horizontal correlation of OE and RE"

10) Line 113: "Around $10^3$ ..." surely the authors mean "at $10^3$ or below"?

**Answer:** We rephrased it.

11) Line 117: The training period concentrates on heavy thunderstorms. Somewhere, the authors should discuss the applicability of their results to other periods, such as wintertime cyclonic systems.

**Answer:** We added in the conclusion "Results presented here are based on the convective period in the summertime. The applicability of those results to other periods such as wintertime cyclonic systems needs further investigation. However, some studies have been done by the other centers. For instance for the Met Office UKV model with the 3D-VAR scheme, the estimated OE statistics (based on Desroziers' method) for radial wind are qualitatively similar to those in the summertime (Waller et al. 2016c), and for reflectivity, Kouroupaki 2019 shows that the estimated standard deviations of the OE in Winter are larger than in Summer and that they increase with reflectivity values".

12) Line 131-133: "It is noticed that the variations of standard deviations are very comparable to the simulated reflectivities of the (high resolution) model run in Fig. 4c, indicating a systematic error that is proportional to the true value". This does not make sense: Figure 4a and 4c do not look very similar, there is no strong similarity between the two. Also it is not supported why this should be explained by a systematic error.

**Answer:** We rephrased the text as "Standard deviations increase in the first few hundred meters and then slightly decrease for the next few hundred meters before increase to a local 160 maximum at around 3 km. Above 3 km, standard deviations decrease till 5 km and then increase to the top. The variations of simulated reflectivities of the model run exhibit a similar pattern although the decrease between 2 and 6 km is sharper. Overall, it can be said that standard deviations are approximately proportional to observed values."

13) Line 133-134: Comparison of Figure 4 to Figure 5 is not so helpful because the former is based on the $< 5$ dBZ sample, and the latter is based on all data.

**Answer:** We added Figures for reflectivity data $\geq 0$ dBZ.

14) Line 208: "too big" - define?

**Answer:** We rephrased text as "a first guess check is carried out (i.e., the innovation of the deterministic run must be smaller than three times of the standard deviation of the innovation)".

15) Line 312-314: "the model produces too much ice" - this does not appear to have been much discussed or supported in the preceding text.

**Answer:** Thank you for noticing. It was a mistake, actually we meant graupel instead of ice. It is mentioned in Line 299-301.

16) Line 314-330: "this is mainly due to the application of scaling factor...". Unless I missed it, this scaling factor has not already been discussed in the text, and its effect on the Desroziers-based observation error estimate has not been established.

**Answer:** The explanation of the scaling factor is given in Line 230-233 and its effects are mentioned in Line 270-271.

---

## Author Comment (AC2)

**To Reviewer 2**

Zeng et al present an interesting paper about observation error statistics for radar reflectivity and Doppler radar wind measurements. Their study includes an estimation of the covariances arising from the error due to unresolved scales based on model data, and estimation of the full observation observation error covariances using the Desroziers et al (2005) assimilation diagnostics method.

**Answer: Thank you very much for your kind acknowledgment of our work.**

While the results for the Desroziers et al (2005) diagnosis of the Doppler radar winds are a little incremental (these have been published for a previous version of the DWD KENDA assimilation system by Waller et al, 2019), their comparison with the model-derived representation error statistics provided some fresh ideas. Furthermore, diagnosed estimates of the radar reflectivity error covariances have not been published in the mainstream literature before. I found the paper to be lacking a little background information, which might provide a deeper understanding of the results presented. I also had some minor questions about the experimental methods and results. I believe that these can be addressed very straightforwardly by the authors. My specific comments follow:

**Answer:** We added now following text in the introduction: In the present work, we use the Desroziers method to explore characteristics of the OE for radial wind and reflectivity in the operational ICON-KENDA system of the DWD. It is the first application of radar data assimilation using this framework (a similar study has been done by Waller et al. 2019 but for the COSMO-KENDA system and only for the radial wind). To authors' knowledge, it is also the first in-depth attempt to investigate the OE statistics (variances and correlations) of reflectivity data. However, the estimated OE statistics embraces contributions from the IE, FE and RE and it is not clear how much an individual error contributes. To approximate the RE, we assume that a high resolution model is the truth and we regard model equivalence of radar data calculated from the truth as observations (e.g., Waller et al 2014, Waller et al 2021) and evaluate the statistics from a set of samples of differences between observations and model equivalence of the low resolution model run, which can then be compared with the OE statistics estimated by the Desroziers method.

1. There was very little review provided of the expected sources of uncertainty for the observations. I believe that giving this background could provide more insight in the results. For instance:

(a) There is previous literature noting the dependence of the reflectivity error variability on the reflectivity value e.g.,

Doviak, R. J., and D. S. Zrnic, 1993: Doppler Radar and Weather Observations. 2nd ed. Academic Press, 562 pp.

Xue, M., Jung, Y., and Zhang, G. (2007). Error modeling of simulated reflectivity observations for ensemble Kalman filter assimilation of convective storms. Geophysical research letters, 34(10).

**Answer:** We added the references. The instrumental error of radar reflectivity observations is proportional to the measured values (Doviak and Zrnic, 1993 and Xue et al. 2007).

(b) Waller et al (2019) pointed out the contributions to the Doppler radar wind observation errors from the DWD superobbing scheme.

**Answer:** We mentioned "As noted by Waller et al. 2019, this superobbing technique may create error correlations since the same raw observations may be accounted for in neighboring superobservations". However, we do not fully agree with their interpretation of effects of superobbing on error correlations. The correlations arise since the wedges of the neighboring superobbing points overlap. In case of large size of overlap, strong correlations arise. To our understanding, there are several factors influencing the size of overlap for the the DWD superobbing scheme: First, the range or height to radar stations: on one hand, the closer the superobbing point is, the broader is the width of azimuth for supperobbing (see Fig. A7), which results in a larger overlap for neighboring superobbing points. On the another hand, two neighbouring superobbing points are further away when they are closer to radar stations, which results in a smaller overlap. Therefore, the size of overlap depends on these two competing factors. In addition, the size of the overlap also depends on the elevation. If the neighboring superobbing points have the same distance to the radar station, they are further away from each other if they are from higher elevations than from lower elevations, therefore, smaller overlap for higher elevations. The dependency of the size of overlap on the distance is not straightforward and for the moment we can not draw conclusion. However, we did another experiment with superobbing resolution of 10 km, which results in longer correlation length scales. This is probably due to larger overlapp of superobbing wedges. We mentioned this in the text.

(c) How might reflectivity attenuation (in a heavy storm) affect the results?

**Answer:** Reflectivity attenuation is an important error source and can have considerable impacts on the statistics of the OE, but this requires rigourous studies that can be done in the future.

2. A little more information about the form of the operator T is needed. The reader should not need to access Zeng et al (2019) in order to understand what this operator does.

**Answer:** T is the interpolation operator and the one used in this work is the iconremap utility from the DWD ICON Tools (Prill 2014). We added "using the the iconremap utility from the DWD ICON Tools (Prill 2014)," to Line 145.

Prill, F., 2014:DWD ICON Tools Documentation. Deutscher Wetterdienst (DWD),dwdicontools/doc/icontoolsdoc.pdf.

3. Localization: Waller et al (2017) pointed out an issue using the Desroziers et al (2005) method together with localization, and provided some criteria to establish which observation pairs can be used in the calculation. Was this method followed in this paper? What is the localization radius used in the experiments?

J.A. Waller, S. L. Dance, and N. K. Nichols, 2017: On diagnosing observationerror statistics with local ensemble data assimilation. Quart. J. Roy. Meteor. Soc., 143, 2677-2686, https://doi.org/10.1002/qj.3117.

**Answer:** The R-localization is applied in the LETKF, which is a type of domain localization as used in Waller et al. 2017. It is the same method used in Waller et al. 2019. The information on the localization radius is given in Line 236-237.

4. There is a further recent publication (Waller et al, 2021) providing modelbased estimates of errors due to unresolved scales that are more appropriate for convection-permitting lengthscales than the earlier 2014 paper that is cited. The Waller et al (2021) statistics for zonal and meridional wind standard deviations decrease with height, as is also largely reflected by the black lines in the relevant panels in Figure 5 in this paper (above the first few km). However, the opposite holds for the radial winds in Figure 7 (i.e. the error standard deviations for the radial winds increase with height). I did not understand the explanation for this difference in the paper.

Waller, J.A., Dance, S.L. and Lean, H.W. (2021), Evaluating errors due to unresolved scales in convection permitting numerical weather prediction. Q J R Meteorol Soc. Accepted. doi:10.1002/qj.4043

**Answer:** There are several possible explanation for this difference. First, Waller et al, 2021 discussed the RE of horizontal wind and this article focuses on the radial wind. These are two different types of measurements. The latter one also contains information about the vertical wind. Therefore, the results of Waller et al, 2021 do not necessarily hold here. Second, intuitively it is not surprising that the standard deviations of errors increase with the height since the wind speed increase with the height. Third, the weather conditions can be also a factor. The study period in this article is with many thunderstorms and deep convective clouds. Fourth, the RE of lower elevation increase faster with the height in our study, indicating that the RE of horizontal wind is dominating with the height. This can be due to the fact that anvil regions are approached where divergent convective outflows occur and winds move in different directions and slight spacial shifts of cells in simulations can lead to large errors, therefore, the standard deviations increase.

5. L18 and L40 are a little out of date. The current operational system at the UK Met Office uses 4D-Var. Some more up to date references:

Milan, M, Macpherson, B, Tubbs, R, et al. Hourly 4D-Var in the Met Office UKV operational forecast model. Q J R Meteorol Soc. 2020; 146: 1281-1301. https://doi.org/10.1002/qj.3737

Hawkness-Smith, LD, Simonin, D. Radar reflectivity assimilation using hourly cycling 4D-Var in the Met Office Unified Model. Q J R Meteorol Soc. 2021;

**1516-1538. https://doi.org/10.1002/qj.3977**

**Answer:** Thank you. We added these references.**

6. L44 The JMA have also used the Desroziers et al (2005) method with radar data and this should be noted. See for example,

Fujita, T., Seko, H., Kawabata, T., Ikuta, Y., Sawada, K., Hotta, D. and Kunii, M. (2020) Variational Data Assimilation with Spatial and Temporal Observation Error Correlations of Doppler Radar Radial Winds. Research activities in Earth system modelling. Working Group on Numerical Experimentation. Report No. 50.WCRP Report No.12/2020. WMO, Geneva.

**Answer:** We added the reference.**

7. A short study estimating reflectivity variances using Desroziers et al (2005) was carried out by a Masters student using UK Met Office trial data. Some comparison could be made with these results.

Kouroupaki, V. (2019). Investigating radar reflectivity uncertainty in data assimilation for high impact weather prediction. MSc Thesis. University of Reading, UK.

**Answer:** We added the reference.**

8. Fig 2. Is not referred to in the text until p14. It would be better to refer to this figure earlier, in section 3.

**Answer: We switched the order of Figures 2 and 3.**

9. Fig 3 caption - what is meant by "scratch"?

Answer: We changed "Scratch" to "Illustration".

10. Line 113 what do you mean by "statistically insignificant" here? How many samples are needed for reliable estimation?

**Answer:** We rephrased "As Waller et al. 2019, if the numbers of samples available for estimation are too small (e.g., < 1000), the estimated standard deviation and correlations might be considerably contaminated by the sampling error and therefore are not reliable".

11. Section 4.1 Do these RE estimation experiments include superobbing?

**Answer:** No superobbing has been applied in the RE estimation. We added this in the text. This is worth testing in the future

12. Section 4.1 Does the representation error exhibit a bias? At line 132 "systematic error" is mentioned, but the plots only show standard deviations,

so cannot give an indication of biases.

**Answer:** We calculated the means of the representation error for both reflectivity and radial wind, which are very close to zero, i.e., no bias. Therefore, we removed the term of "systematic error".

13. Fig 4a Why is there a sharp gradient at very low levels?

**Answer:** The sharp gradient at very low levels may correspond to increasing reflectivity values at these levels as shown in Fig 4c. Furthermore, it can be due to representation error of orography (Waller et al. 2021) or due to turbulence at the boundary layer, but it is not certain.

14. Reflectivity correlations: For the standard deviations a clear dependence on reflectivity value was shown. Might this also apply to the correlation lengths? Would it be appropriate to produce correlation plots separated by reflectivity value rather than beam elevation?

**Answer:** It is an interesting point but it is technically complicated to implement this. We may investigate this in the future.

15. Section 4.2 Please could you clarify what happens to the "dry" observations (zero/small reflectivity)? Are they assimilated? In the text there is some mention of "no reflectivity data" (line 193, 222) but it wasn't clear to me what this referred to.

**Answer:** Reflectivities with values smaller than 0 dBZ are set to 0 dBZ and treated as no reflectivity data and are assimilated. We discussed this in Line 94-103.

16. Section 4.2: Are the O-As and O-Bs used for calculating the Desroziers et al (2005) diagnostic unbiased? If not, is the bias subtracted before computing the covariances?

Answer: We now subtract the means from O-As and O-Bs.

17. Fig 11 (and later figures). The right hand panel (no of samples) displays a zig-zag pattern (most obvious for purple and blue lines). Why is this?

**Answer:** The reason for this is not clear (probably due to the superobbing methods) but it does not jeopardize the interpretation of results.

18. Line 255 you explain a difference in size of standard deviation compared with previous work due to a scaling factor in R. Can you explain this more clearly? Is this to do with the deficiencies of the diagnosis technique?

**Answer:** First of all, the scaling factor is introduced because the IE of radial wind measurements are usually large if onsite reflectivities are too small (Line 231-234), not due to the deficiencies of the Desroziers method. The scaling factor inflates R of radial wind where small reflectivities (i.e., between 0 and 10

dBZ) are observed, which makes the estimated standard deviation larger. However, it is recognized in practice that the Desroziers method tends to produce too small variances (Weston et al., 2014; Bormann et al., 2016), the introduction of the scaling factor also compensates the deficiency of the method (see Line 371-372).

19. The earlier paper on Doppler wind error estimation with the KENDA system (Waller et al, 2019) emphasizes the role of the superobbing procedure in generating error correlations. Is the superobbing procedure used here the same? Do some of the error correlations arising here stem from the overlapping superobbing wedges?

**Answer:** The same superobbing method is used here. We did another experiment with superobbing resolution of 10 km, which results in longer correlation length scales. This is probably due to larger overlapp of superobbing wedges. We mentioned this in the text.

20. Figure 9 is not referred to in the main text. It does not seem useful to include if it is not referred to.

Answer: We moved the Figure to Appendix.

21. Figure 10 could be cut as it does not tell us very much.

**Answer:** Fig. 10 is to demonstrate that superobbed observations are evenly distributed in horizontal. We put the figure in Appendix.

22. I feel that the paper would benefit from being a bit more selective about which figure panels to show to make the relevant points e.g., is it necessary to show correlations for every elevation? Or could the key points be made from one or two elevations, and the rest of the figures put in supplementary material.

**Answer:** We have selected five elevations from ten after rigorous consideration, with which dependency of important features of statistics can be well seen. Since we see the elevation  $8^{\circ}$  behaviors similarly as  $5.5^{\circ}$  and the numbers of samples available are not always sufficient, we removed the elevation  $8^{\circ}$ .

**Typos and Small corrections**

Line 46 "authors's"

Line 70 and throughout "setup" is rather informal.

Line 80 Parentheses needed around Waller et al (2016b)

Line 101 EMVORDAO

Line 137 grauple  $% \left( {{{\rm{T}}_{{\rm{T}}}}_{{\rm{T}}}} \right)$

Answer: Done.

---

## Author Response (AR2)

**To Reviewer 1**

The authors have done a good job of revising the paper. There are two small outstanding issues that I would like to raise again as they were partially but not fully addressed in the revision. This may be my fault for not being clear enough in my comments.

1. Thanks to the revision, I have better understood what is meant by "no reflectivity data". However, I do find the nomenclature "no reflectivity data" to be misleading. It sounds like missing data, rather than existing data with small or zero dBZ value.

**Answer:** The wording of "no reflectivity data" is quite commonly used in the radar data assimilation community. However, considering that this may be misleading, we changed it to "clear-air reflectivity data"

2. Localization and Desroziers et al (2005) calculations. I think I did not make my point clearly enough. As noted by Waller et al (2019) "The diagnostic in Desroziers et al. (2005) is derived assuming that the analysis is calculated using minimum variance linear statistical estimation. However, the diagnostic no longer provides a correct estimate of the observation error covariance matrix if local ensemble data assimilation is used to calculate the analysis. Using a modified version of the diagnostic it is possible to recover some of the observation error statistics. Waller et al. (2017) show that the diagnostic can be used to estimate the error correlations between two observations if the observation operator that determines the model equivalent of observation $y_i$ acts only on states that have been updated using the observation $y_j$. Otherwise, the error correlation cannot be estimated using the diagnostic."

As this paper follows a similar methodology to Waller et al (2019) it is probably the case that the calculations do not suffer from this issue. However, I think the authors should verify this and mention it in their paper.

**Answer:** We added "Furthermore, it is noted in Waller et al. 2017 that for the local ensemble data assimilation scheme the error correlation between two observations $y_i$ and $y_j$ estimated by the Desroziers method is correct if the observation operator applied to calculate the model counterpart of $y_i$ acts only on states updated by $y_j$, however, the LETKF does not seem to suffer strongly from this issue as shown in Waller et al. 2019".

**Typos**

L33 "paramters (?)"

L184 "e.g. ?"

L207 "spacial"

**Answer:** Done.